



# Predicting mangrove forest dynamics across a soil salinity gradient using an individual-based vegetation model linked with plant hydraulics

Masaya Yoshikai[1], Takashi Nakamura[1], Rempei Suwa[2], Sahadev Sharma[3], Rene Rollon[4], Jun Yasuoka[5],
Ryohei Egawa[5], Kazuo Nadaoka[1]

[1]School of Environment and Society, Tokyo Institute of Technology, Tokyo, 152-8552, Japan
[2]Forestry Division, Japan International Research Center for Agricultural Sciences (JIRCAS), Ibaraki, 305-8686, Japan
[3]Institute of Ocean and Earth Sciences, Universiti Malaya, Kuala Lumpur, 50603, Malaysia
[4]Institute of Environmental Science & Meteorology, College of Science, University of the Philippines, Diliman, Quezon
City, 1001, Philippines
[5]Graduate School of Information Science and Engineering, Tokyo Institute of Technology, Tokyo, 152-8552, Japan

*Correspondence to*: Masaya Yoshikai (yoshikai.m.aa@m.titech.ac.jp)

**Abstract.** In mangrove forests, soil salinity is one of the most significant environmental factors determining mangrove forest distribution and productivity as it limits plant water uptake and carbon gain. However, salinity control on mangrove
productivity through plant hydraulics has not been investigated by existing mangrove models. Thus, we present a new individual-based model linked with plant hydraulics to incorporate physiological characterization of mangrove growth under salt stress. Plant hydraulics was associated with mangroves nutrient uptake and biomass allocation apart from water flux and carbon gain. The developed model was performed for two-coexisting species of *Rhizophora stylosa* and *Bruguiera gymnorrhiza* in a subtropical mangrove forest in Japan. The model predicted that the productivity of both species was
affected by soil salinity through downregulation of stomatal conductance, while *B. gymnorrhiza* trees grow faster and suppress the growth of *R. stylosa* trees by shading that resulted in a *B.gymnorrhiza*-dominated forest under low soil salinity conditions (< 28‰). Alternatively, the increase in soil salinity significantly reduced the productivity of *B. gymnorrhiza* compared to *R. stylosa*, leading to an increase in biomass of *R. stylosa* despite the enhanced salt stress (> 30‰). These predicted patterns in forest structures across soil salinity gradient remarkably agreed with field data, highlighting the control
of salinity on productivity and tree competition as factors that shape the mangrove forest structures. The model reproducibility of forest structures was also supported by the predicted self-thinning processes, which likewise agreed with field data. In addition, the mangroves morphological adjustment to increasing soil salinity – by decreasing transpiration and increasing hydraulic conductance – was reasonably predicted. Aside from the soil salinity, seasonal dynamics in atmospheric variables (solar radiation and temperature) was highlighted as factors influencing mangrove productivity in a subtropical
region. The physiological principle-based improved model has the potential to be extended to other mangrove forests in various environmental settings, thus contributing to a better understanding of mangrove dynamics under future global climate change.



## 1 Introduction

Mangrove forests grow in intertidal zones in tropical and subtropical regions (Giri et al., 2011) and store a large
amount of carbon (C) especially in their soil, commonly referred to as "blue carbon". It has roughly four times higher
ecosystem-scale carbon stock than other forest ecosystems (Donato et al., 2011), characterizing them as globally important C
sinks (Mcleod et al., 2011; Alongi, 2014; Taillardat et al., 2018; Sharma et al. 2020), therefore playing an important role in
climate change mitigation. However, mangrove forests have declined worldwide; at least 35% of the mangrove forests had
disappeared in the 1980s and 1990s predominantly because of deforestation due to conversion to aquaculture ponds, rice
fields, urban development and palm oil plantations (Friess et al., 2019). Deforestation has been continuing until now
particularly in Southeast Asia with a recent estimate of mangrove loss rates between 0.11%–0.70% (Friess et al., 2019,
2020). The loss of mangrove soil C through mineralization following deforestation has been of concern as a source of carbon
emission to the atmosphere in addition to the loss of C sequestration capacity (Atwood et al., 2017; Sharma et al. 2020;
Adame et al., 2021). To facilitate effective mangrove conservation, management, and restoration, a better understanding of C
sequestration rates and the soil C dynamics, hence mangrove blue C dynamics, under different environmental conditions and
climate change are urgently needed.

While the mangrove soil C dynamics are complex and involve physical, biogeochemical, and ecological processes
(Kristensen et al., 2008; Alongi, 2014; Bukoski et al. 2020) that still remains poorly understood, one of the most important
variables determining soil C dynamics may be related to mangrove productivity. Mangroves supply their products, such as
leaf litter and dead roots to the soil C pool (Kristensen et al., 2008; Alongi, 2014; Ouyang et al., 2017) which is closely
related to forest structural variables such as canopy height and above-ground biomass (AGB) (Saenger and Snedaker, 1993;
Komiyama et al., 2008). Such autochthonous C accounts for a significant amount of total soil C in mangrove forests (Xiong
et al., 2018; Sasmito et al., 2020). Therefore, the aim of this study is to successfully quantify and predict the biomass
dynamics and growth processes of mangroves in different environmental conditions. These results would take a step forward
in our understanding of mangrove C sequestration rate and soil C dynamics.

Although data and insights on mangrove AGB distributions in relation to environmental variables have recently
increased (Simard et al., 2019; Rovai et al., 2015, 2021), there is still no established way to predict the dynamics of
mangrove AGB in the changing environmental conditions. Generally, ecosystem's response to environmental variables is
nonlinear, and biomass dynamics is cumulatively affected by nonlinear response. Therefore, predicting the effect of one
environmental variable on mangrove biomass dynamics is difficult if based only from the monitoring data on mangroves
biomass, which are exposed to the effects of multiple environmental variables. This makes the assessment of environmental
impacts on mangrove biomass dynamics challenging if datasets from only the field-based monitoring approach are used.

The dynamic vegetation model (DVM) simulates vegetation or forest growth based on physiological principles that
includes processes such as tree competition, establishment, and mortality (Fisher et al., 2017). This model could be a way to
overcome the limitation of field-based approach and predict mangrove biomass dynamics under multiple environmental





variables. Various DVMs (e.g., big-leaf, cohort-based, individual-based) have been developed mainly for terrestrial ecosystems and have successfully reproduced the dynamics of various forests in the temperate, tropical, and boreal regions (Fisher et al., 2017). Recently, DVMs have advanced in physiological expression of stomatal conductance under water stress, by incorporating a plant hydraulic model that explicitly solves plant water flux (Bonan et al., 2014; Xu et al., 2016; Li

et al., 2021). Recent studies also identified plant hydraulics as a critical factor that determines the plants' biomass allocation pattern to leaves, stem, and roots (Magnai et al., 2000; Trugman et al., 2019b; Portkay et al., 2021), the variations of which could drive a significant variation in plant productivity (Trugman et al., 2019a).

In mangrove forests, the salt in soil porewater (soil salinity) is one of the significant environmental factors that determine the mangroves distribution, productivity, structure, and zonation pattern (Ball and Farquhar, 1984; Clough and

Sim, 1989; Sobrado, 2000; Ball, 2002; Suarez and Medina, 2005; Suwa et al., 2009; Barr et al., 2013; Nguyen et al., 2015). Therefore, it is essential to properly represent the effects of soil salinity on mangrove growth considering species differences in the tolerance of salinity in order to accurately predict the mangrove biomass dynamics. Soil salinity imposes highly negative water potential in the substrate, making the water acquisition energetically challenging for plants, which acts in a similar way to water stress (Reef and Lovelock, 2015). With this perspective, the theoretical works of Perri et al. (2017,

2019) demonstrated the importance of considering the plant hydraulics for predicting the photosynthetic and transpiration rates under salt stress. However, although there are several individual-based DVMs for mangroves (e.g., FORMAN by Chen and Twilley, 1998, Kiwi by Berger and Hildenbrandt, 2000, mesoFON by Grueters et al., 2014, and BETTINA by Peters et al., 2014), no model yet has considered salinity control role in photosynthesis and transpiration through plant hydraulics, suggesting a room for improvement in the physiological representation of the mangrove biomass dynamics under the impacts

of soil salinity. It is expected that the nutrient uptake rate is also affected by soil salinity through the regulated transpiration rate (Simunek and Hopmans, 2009), making nutrient availability as one of the key factors controlling mangrove growth especially under high soil salinity conditions (Lovelock et al., 2004, 2006a, 2006b; Feller et al., 2007; Reef et al., 2010). Nonetheless, the modeling studies have not explicitly considered the role of nutrient uptake in mangrove growth.

Here we hypothesized that the individual-based DVM incorporating plant hydraulic traits can reasonably predict

mangrove biomass, structure, and species zonation pattern across a soil salinity gradient without empirical expression of the soil salinity influence on mangrove productivity. Such model would advance the understanding of mangrove biomass dynamics under multiple environmental stresses, which ultimately influence the mangrove soil carbon dynamics. To test the hypothesis and contribute to the improvement of the physiological representation of mangrove growth specifically under soil salinity impacts, we developed a new individual-based DVM for the mangrove forest. The developed model is based on a

terrestrial individual-based DVM – the SEIB-DGVM (Spatially-Explicit Individual-Based Dynamic Global Vegetation Model, Sato et al., 2007). We added a plant hydraulic model to SEIB-DGVM and coupled it with the photosynthetic model to consider the impacts of soil salinity on the mangrove water uptake and carbon gain. We also explicitly considered the role of nutrient uptake on biomass dynamics. Furthermore, a novel biomass allocation scheme linked with plant hydraulics and resource uptake rate was introduced as the mangroves strategy to cope with salt stress and enhance the rate of production.



We tested the developed model and determined the reproducibility of forest structures (e.g., species composition, biomass) in a subtropical mangrove forest in Japan with two coexisting species (*Rhizophora stylosa* and *Bruguiera gymnorrhiza*).

## 2 Materials and Methods

### 2.1 Study sites

Our study site for the model application is an estuarine mangrove of the Fukido River (Fukido mangrove forest) in
Ishigaki Island, Japan (Fig. 1, 24° 20' S, 124° 15' E). The site is characterized as a subtropical region. According to the climatological normal data obtained by the Japan Meteorological Agency, the annual-mean air temperature is 24.5 ℃, with a monthly average of 29.6 ℃ in July and 18.9 ℃ in January (see also Fig. 4). The mean monthly precipitation is 142 mm in July and 135 mm in January. Four small rivers (R1 – R4) flow into the Fukido mangrove forest, while the river R2 has two outlets (Fig. 1c). The mean discharge rates of the rivers in October 2012 were less than 0.03 $m^3$ $s^{-1}$ for R1, R3, and R4 and
around 0.05 $m^3$ $s^{-1}$ for R2 (Mori et al., unpublished data). The tide is semi-diurnal with the highest and lowest amplitude of 1.8 m and 0.8 m, respectively (Egawa et al., 2021).

The site is vegetated by two species, *R. stylosa* and *B. gymnorrhiza*. The trees of *R. stylosa* dominated the sea-ward zone, especially areas close to the river mouth (Fig. 1c) while *B. gymnorrhiza* dominated the landward zone. The species *R. stylosa* is classified as a relatively salt-tolerant species while *B. gymnorrhiza* is classified as a less salt-tolerant but shade-
tolerant species (Putz and Chan, 1986; Sharma et al. 2012; Reef et al., 2015). According to Ohtsuka et al. (2019), the Fukido mangrove forest is a mature and intact mangrove forest designated as natural protection area by Ishigaki City, where distinct disturbances to the mangroves have not occurred since at least 1977 based on aerial photograph analysis.





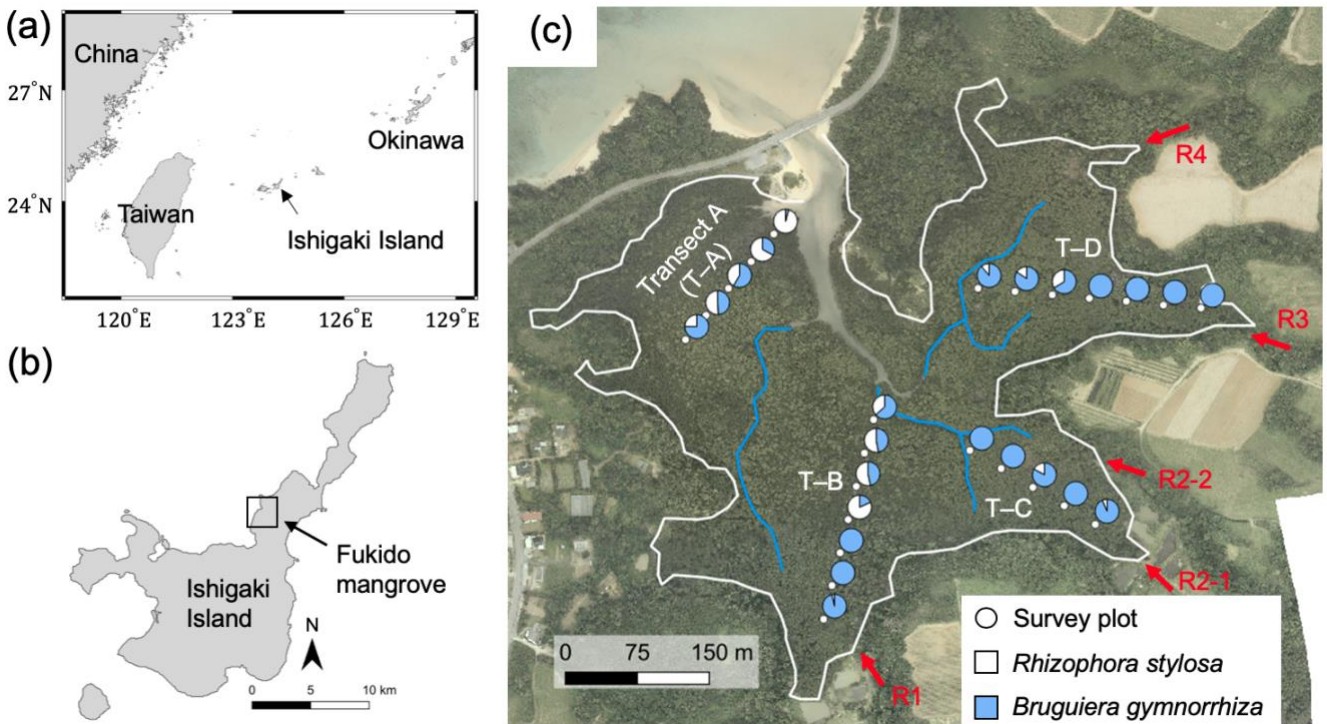

Figure 1. (a) Location of Ishigaki Island, (b) location and (c) aerial photo of the study site – Fukido mangrove forest. The
white line in panel (c) indicates the boundary of mangroves and other land covers where mangroves are assumed to inhabit
the areas of elevation < 1.0 m + mean sea level, which was delineated based on a LiDAR-derived digital elevation model
(DEM). The blue lines indicate small creeks. The circular makers indicate survey plots' locations along with four transects
(T–A to T–D), while the pie charts indicate species composition in each plot. The red arrows indicate outlets of rivers
flowing into the mangrove forest (R1 to R4). The aerial photo and DEM products were obtained from Asia Air Survey Co.
Ltd., Japan. Shorelines are from GSHHG.

## 2.2 Field data collection

We used the tree census data of the Fukido mangrove forest shown in Suwa et al. (2021) to assess model
performance. The tree census data were collected from the survey plots established along four transects (T–A, T–B, T–C,
and T–D), shown in Fig. 1c. The details of the survey protocol are described in Suwa et al. (2021). The stem biomass of
individual trees ($M_S$, g) was estimated from a common mangrove allometric equation proposed by Komiyama et al. (2005),
which was validated with various mangrove species:

$$M_S = 70\rho[(\text{DBH}/100)^2 H]^{0.931} \tag{1}$$

where $\rho$ is the wood density (g cm$^{-3}$), DBH is the stem diameter at breast height (m) divided by 100 for the unit conversion
from meter to centimeter, and $H$ is the tree height (m). However, tree height data were occasionally absent at some plots,





especially along T–C and T–D, and in such cases, the tree height was estimated using a DBH-*H* allometric relationship (Supporting Information Fig. S1a and b). The AGB at each plot (Mg ha$^{-1}$) was then calculated from the estimated stem biomass.

The crown diameter was also measured for some selected trees, besides the data shown in Suwa et al. (2021). The trees for crown measurement were randomly selected at each transect, the diameters parallel and perpendicular to the

transect line were measured for each tree, and the crown diameter ($D_{crown}$, m) was represented by the average of the values from the two directions. Totally, crowns of 81 trees of *R. stylosa* and 103 trees of *B. gymnorrhiza* were measured (Supplementary materials Fig. S1 c and d).

Soil salinity and porewater dissolved inorganic nitrogen concentration (DIN) were also measured at each plot as environmental drivers of mangrove production. Soil samples were collected by inserting a PVC pipe into the soil at each

plot, and soil porewater was extracted from the surface 10 cm soil sample. The porewater samples were kept frozen and brought to the laboratory for analysis. Salinity of the porewater (soil salinity) was measured using a salinity meter (PAL-SALT, ATAGO Co. Ltd., Japan) while DIN concentrations were measured using a QuAAtro 2-HR (SEAL Analytical Ltd., Germany and BLTEC K.K., Japan). These measurements were conducted from August to September 2013. The summary of the environmental and vegetation variables at each plot is provided in Table S1.

**2.3 Model description**

The mangrove growth model was formulated based on an individual-based model, SEIB-DGVM (Sato et al., 2007). The forest dynamics was represented by a 30 m × 30 m computational domain. In this domain, the irradiance distribution, tree establishment, death, and changes in plant morphology subsequent to growth were simulated (Sato et al., 2007). A feature of SEIB-DGVM is that it explicitly solves the effects of shading by neighboring trees on the light acquisition. The

SEIB-DGVM thus provides the advantage in describing tree competition for light more than the other types of DVMs such as big-leaf or cohort-based models (Fisher et al., 2017). In SEIB-DGVM, the crown of each tree is represented by a cylindrical-shaped object divided by 0.1 m-thick crown layers to account for the within-crown vertical variability in irradiance distribution. It is assumed that leaf biomass is evenly distributed in the crown layers.

Originally, the SEIB-DGVM defines four biomass pools – leaf, trunk, and fine root, and stock (non-structural

storage pool); the trunk includes both the above-ground stem and the below-ground coarse root (Sato et al., 2007). In this study, we considered the stem and coarse root separately to explicitly consider the role of coarse root turnover in the biomass dynamics (Castaneda-Moya et al., 2011; Adame et al., 2014). Additionally, we also added a new biomass pool – the above-ground root, especially for *Rhizophora* species whose above-ground root, or "prop root", could account for nearly 60% of their AGB (Nishino et al., 2015; Vinh et al., 2019).

The original SEIB-DGVM does not have a plant hydraulic module and the effects of soil water on stomatal conductance were empirically parameterized. It also does not account for plant nutrient uptake; thus, the plant growth depends solely on photosynthesis. The biomass allocation is modeled based on scaling law (Trugman et al., 2019a). In this




study, these processes that control plant growth were almost entirely modified to describe mangrove growth under salt stress
(Fig. 2). The following sections explain the modification of the SEIB-DGVM for this study related to plant hydraulics. Other
modifications to the SEIB-DGVM are summarized in Note S3–4.

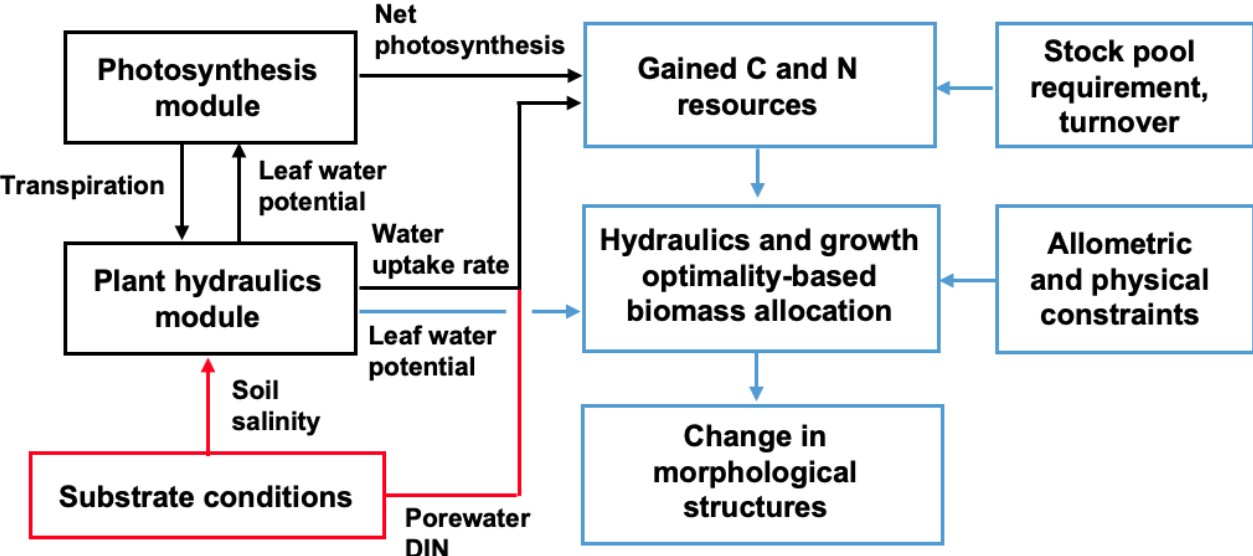

Figure 2. The model framework newly added to SEIB-DGVM for describing mangrove growth. The red box and arrows
indicate the substrate conditions given in the model. The black boxes and arrows indicate processes computed in the hourly
time steps while the blue ones are for the daily time step.

### 2.3.1 Inclusion of plant hydraulic module

The plant hydraulic module implemented in this study is primarily based on the model developed by Xu et al.
(2016) in a soil-plant-atmosphere continuum scheme. Here we describe essential processes in the plant hydraulic module
which will be related to the new biomass allocation model in the next section.

The plant water uptake rate ($\approx$ sap flow rate ($J_{\text{sap}}$, kg H$_2$O tree$^{-1}$ s$^{-1}$)) is calculated as

$$J_{\text{sap}} = \frac{\Psi_{\text{s}} - \Psi_{\text{l}} - \Psi_{\text{h}}}{R_{\text{whole}}} \qquad (2)$$

where $R_{\text{whole}}$ is the whole-plant hydraulic resistance (MPa s tree kg$^{-1}$ H$_2$O) and $\Psi_{\text{s}}$ and $\Psi_{\text{l}}$ are the soil and leaf water potential
(MPa), respectively; the $\Psi_{\text{h}} = \rho_{\text{w}}gH10^{-6}$, which is the gravitational water potential drop from the ground to the crown (MPa),
where $\rho_{\text{w}}$ is the water density (kg m$^{-3}$) and $g$ is the gravitational acceleration (m s$^{-2}$). The parameter $\Psi_{\text{s}}$ can be expressed as a
sum of the matric potential and osmotic potential ($\Psi_{\pi}$, MPa). The parameter $\Psi_{\pi}$ can be expressed as the difference in the
osmotic potential between the soil and plant, which is linearly related to soil salinity and the partial uptake of the salt by
mangroves represented by the salt filtration efficiency, $\varepsilon$ (fraction) (Perri et al., 2017). Alternatively, the matric potential is





negligibly small compared to $\Psi_\pi$ in mangrove forests where the soil is usually water-saturated due to frequent tidal flooding (Perri et al., 2017). The parameter $R_{whole}$ can be expressed as the sum of the root to stem hydraulic resistance ($R_{root}$) and the stem to leaf hydraulic resistance ($R_{stem}$), both expressed in (MPa s tree kg$^{-1}$ H$_2$O). The parameter $R_{root}$ is given by:

$$R_{\text{root}} = \frac{R_r}{M_{FR}} \tag{3}$$

where $R_r$ is the fine root hydraulic resistance per unit biomass (MPa s g kg$^{-1}$ H$_2$O) and $M_{FR}$ is the fine root biomass (g tree$^{-1}$). The parameter $R_{stem}$ is given by:

$$R_{\text{stem}} = \frac{a_1 H}{K_{sap} A_{sap}} \tag{4}$$

where, $a_1$ is the correction factor for tree height ($H$) to water path length, $K_{sap}$ is the stem hydraulic conductivity per unit
sapwood area (kg H$_2$O m m$^{-2}$ sapwood s$^{-1}$ MPa$^{-1}$), and $A_{sap}$ is the sapwood area of a tree (m$^2$ sapwood tree$^{-1}$), which is calculated from the DBH and diameter ratio of the heartwood relative to the entire stem ($\beta_{heart}$, Table 1; Trugman et al., 2019b). The parameter $K_{sap}$ can be expressed as a product of saturated xylem conductivity ($K_{sap,sat}$) and a factor representing the effect of xylem cavitation (Xu et al., 2016):

$$K_{\text{sap}} = K_{\text{sap,sat}} \left(1 + \left(\frac{\Psi_1}{P_{50}}\right)^{a_2}\right)^{-1} \tag{5}$$

where $P_{50}$ (MPa) is the water potential at which 50% of the xylem conductivity is lost and $a_2$ is an empirical parameter (dimensionless). The change in leaf water potential is governed by the equation:

$$\frac{\mathrm{d}\Psi_1}{\mathrm{d}t} = \frac{J - T_{\text{whole}}}{C_p LA} \tag{6}$$

where $T_{whole}$ is the whole-plant transpiration rate (kg H$_2$O tree$^{-1}$ s$^{-1}$), $LA$ is the whole-plant leaf area (m$^2$ leaf tree$^{-1}$), and $C_p$ is the plant capacitance (kg H$_2$O m$^{-2}$ leaf MPa$^{-1}$). The parameter $T_{whole}$ is calculated by vertically integrating the product of the
leaf-level transpiration rate and the leaf area in each crown layer. The leaf-level transpiration and photosynthetic rates and stomatal conductance are calculated using a leaf flux model of Bonan et al. (2014), where the stomatal conductance is estimated from an optimization approach of Cowan and Farquhar (1977) using the marginal water use efficiency ($\lambda = \Delta A_n/\Delta E$, where $\lambda$ is the optimal water use efficiency (WUE), $A_n$ and $E$ are the leaf net photosynthetic rate and the transpiration rate, respectively.) See Note S4 for the detailed calculations of $A_n$ and $E$, and the linkage to $\Psi_1$.

210        The processes for transpiration, photosynthesis, plant water uptake, and change in leaf water potential were computed in hourly time step (Fig. 2). Overall, high salinity increases sensitivity of the leaf water potential to plant transpiration (Eqs. (2), (6)), which in turn may cause stomatal closure even with a low transpiration rate. It also increases the optimal WUE value leading to lower stomatal conductance (Ball and Farquhar, 1984; Clough and Sim, 1989; Barr et al., 2014; Perri et al., 2019), thereby lowering the photosynthetic and transpiration rates.

**2.3.2 Inclusion of hydraulics and growth optimality-based biomass allocation**

        The biomass allocation occurs at the daily time step in the new biomass allocation scheme introduced in this study (Fig. 2). At each time step, four variables were considered for biomass allocation of individual trees – the daily C ($C_{grow}$, g C





tree$^{-1}$ day$^{-1}$) and N (N$_{grow}$, g N tree$^{-1}$ day$^{-1}$) resources that can be used for tree growth, the daily minimum leaf water potential ($\Psi_{l,daymin}$, MPa), and the midday photosynthetically active radiation (PAR) at the crown top (PAR$_{top}$, μmol photon m$^{-2}$ s$^{-1}$).

The C$_{grow}$ and N$_{grow}$ were computed from the daily C and N uptake rates, where N uptake rate was calculated by multiplying the porewater DIN concentration and plant water uptake rate (See Note S5 for the detail). Biomass was allocated according to these variables to optimize the plant hydraulics and enhance the uptake rate of growth-limiting resource (C or N) under the constraints summarized in Table 1. Allometric and physical constraints were considered for $H$ and $D_{crown}$ (Fig. 3a–d, see Note S1 for the derivation of the allometric constraints).

Table 1. Parameters constraining plant morphology, biomass proportion, and stoichiometry. R. s = *R. stylosa*, B. g = *B. gymnorrhiza*.

| Type of constraint | Symbol | Description | Related portion | Units | R. s | B. g | Source |
|---|---|---|---|---|---|---|---|
| Morphological structure | $H_{max}$ | Maximum tree height relative to stem diameter | Tree height | m | a | | Field data |
| | $H_{con}$ | Physical constraint on tree height | Tree height | m | b | | |
| | $D^*_{crown}$ | Maximum crown diameter relative to stem diameter | Crown diameter | m | c | | Field data |
| | $D_{crown,con}$ | Physical constraint on crown diameter | Crown diameter | m | b | | |
| | DBH$_{max}$ | Species-specific maximum stem diameter | Stem diameter | m | 0.25 | 0.45 | Field data |
| | $\beta_{heart}$ | Diameter ratio of heartwood relative to entire stem | Sapwood cross-sectional area | – | 0.15 | 0.15 | Sato et al. (2007) |
| Biomass pool | dLAI$_{max}$ | Maximum leaf area index per 1 m vertical height | Leaf biomass | m$^{-1}$ | 0.8 | 0.8 | Estimated from Clough et al. (1997) |
| | $\beta_{stock}$ | Target C and N in stock pool relative to stem | C and N in stock pool | – | 0.05 | 0.05 | |
| | $\beta_{FR}$ | Target fine root biomass relative to coarse root | Fine root and coarse root biomass | – | 0.2 | 0.2 | Literature survey[d] |
| | $\beta_{AR}$ | Target prop root biomass ratio relative to stem | Above-ground biomass of *Rhizophora* species | – | e | | Yoshikai et al. (2021) |
| Stoichiometry | CN$_l$ | C/N ratio in leaf tissue | Leaf | g C g$^{-1}$ N | 47 | 47 | Tanu et al. (2020) |
| | CN$_w$ | C/N ratio in woody tissue | Stem, above-ground root, coarse root | g C g$^{-1}$ N | 280 | 280 | Alongi (2003), Alongi et al. (2004) |





| | | | | | | |
|---|---|---|---|---|---|---|
| CN$_r$ | C/N ratio in fine root tissue | Fine root | g C g$^{-1}$ N | 103 | 103 | Alongi (2003) |

a. Derived from DBH-$H_{max}$ relationship. See Note S1 and Fig. S1 for details.

b. Computed in the model. See Fig. 3c-d.

c. Derived from DBH-$D^*_{crown}$ relationship. See Note S1 and Fig. S1 for details.

d. Average of values reported in Tamooh et al. (2008), Castañeda-Moya et al. (2011), Adame et al. (2014), Robertson et al. (2016), and Muhammad-Nor et al. (2019).

e. Estimated from prop root allometry in Fukido mangrove forest. See Fig. S3.

Figure 3. Schematics of (a, b) allometric and (c, d) physical constraints on tree height ($H_{max}$, $H_{con}$) and crown diameter

($D^*_{crown}$, $D_{crown,con}$), where the $H_{con}$ and $D_{crown,con}$ in panels (c) and (d) are for the tree with crown filled by yellow color, and





(e) newly added biomass allocation scheme to SEIB-DGVM. See Note S1 for the derivation of allometric constraints from field data.

The parameters $C_{grow}$ and $N_{grow}$ are allocated to the respective biomass pools in a scheme shown in Fig. 3. We applied the concept that plants keep their favorable hydraulic conditions throughout the growth periods by adjusting the
morphological structures (Magnai et al., 2000). In this regard, we introduced a parameter $\Psi_{lk}$ – the critical leaf water potential (MPa) – at which plants aim to maintain their leaf water potential (note that $\Psi_{lk}$ is different from $\Psi_{l,min}$ at which plants close the stomata). It was then considered that when $\Psi_{l,daymin}$ fell below $\Psi_{lk}$, the plant tries to reduce $R_{whole}$ by allocating biomass to either the fine root, or stem, which reduces $R_{whole}$ more effectively (Case 1 and 2 in Fig. 3; note that decreases in $R_{stem}$ and $R_{root}$ were expressed by negative value):

$$M_{FR,t} = M_{FR,t-1} + dM_{FR} \qquad if \ \frac{dR_{root}}{dM_{FR}} < \frac{dR_{stem}}{dM_S} \tag{7a}$$

$$M_{S,t} = M_{S,t-1} + dM_S \qquad if \ \frac{dR_{root}}{dM_{FR}} > \frac{dR_{stem}}{dM_S} \tag{7b}$$

where $M_{FR,t}$ and $M_{S,t}$ are the fine root and stem biomass (g tree$^{-1}$) at time step t (day), and $dM_{FR}$ and $dM_S$ are the daily biomass increment potential of fine root and stem (g tree$^{-1}$ day$^{-1}$), respectively, which are limited by either of $C_{grow}$ and $N_{grow}$ and represented as:

$$dM_{FR} = \frac{1}{C_M} \times min[C_{grow}(1 - F_{gr})(1 - F_{CR,C}), N_{grow}(1 - F_{CR,N})CN_r] \tag{8a}$$

$$dM_S = \frac{1}{C_M} \times min[C_{grow}(1 - F_{gr})(1 - F_{AR}), N_{grow}(1 - F_{AR})CN_w] \tag{8b}$$

where, $C_M$ is the carbon mass per unit dry weight in plant tissue (g C g$^{-1}$ DW), $F_{gr}$ is the growth respiration fraction, $F_{CR,C}$ and $F_{CR,N}$ are the fractions of $C_{grow}$ and $N_{grow}$, respectively, to be allocated to the coarse root to realize $\beta_{FR}$ (target fine root biomass relative to coarse root; Table 1), $F_{AR}$ is a fraction of the resources to be allocated to the above-ground root to realize
$\beta_{AR}$ (target prop root biomass relative to stem; Table 1, also see Fig. S3), and $CN_r$ and $CN_w$ are the CN ratios in fine root and woody tissue (g C g$^{-1}$ N), respectively, that convert the unit of $N_{grow}$ to $C_{grow}$. In Eq. (7), the $dR_{root}/dM_{FR}$ is calculated from Eq. (3), while $dR_{stem}/dM_S$ is calculated from:

$$\frac{dR_{stem}}{dM_S} = \frac{dA_{sap}}{dM_S} \times \frac{dR_{stem}}{dA_{sap}} \tag{9}$$

where $dA_{sap}/dM_S$ is given from Eq. (1) by calculating the increase of DBH with stem biomass increment $dM_S$ without height
growth, and $dR_{stem}/dA_{sap}$ is given from Eqs. (4), (5) where $\Psi_{l,daymin}$ is used in Eq. (5). It should be noted that the variables $\Psi_{l,daymin}$, $C_{grow}$, and $N_{grow}$ change with various factors including atmospheric and substrate variables and tree competition, and no absolute optimal biomass proportion achieves the condition $dR_{root}/dM_{FR} = dR_{stem}/dM_S$ throughout the computational period. Also, due to the different CN ratios in fine root and woody tissues, the increment in stem biomass ($dM_S$) with a unit N resource is greater than that of the fine root biomass ($dM_{FR}$) under N-limited conditions (Eq. (8), Table 1).

Alternatively, if plants are not stressed by the lowered leaf water potential ($\Psi_{l,daymin} > \Psi_{lk}$), the resources are allocated to a plant organ that effectively increases the uptake rate of either C or N, limiting the growth rate. Under N-limited





conditions, plants allocate biomass to the leaves to increase whole-plant transpiration capacity, which increases $N_{gain}$ nearly proportionally (as suggested by Eq. (S22)) (Case 3 in Fig. 3); this is considering that the limited uptake of N is due to the small transpiration rate rather than water uptake regulation by hydraulic resistance. The increase in leaf biomass increases

either $D_{crown}$ and dLAI (leaf area index per 1 m vertical height) depending on the $D_{crown}$ relative to $D^*_{crown}$ and $D_{crown,con}$ (see Note S6 for the details). However, if the increase in leaf biomass is inhibited by $dLAI_{max}$ (maximum dLAI; Table 1) and crown allometry or physical constraint, the resources are allocated to the stem for height growth, which in turn will make a new crown layer and eventually allow further leaf accommodation (Case 4 in Fig. 3). Under a C-limited condition, the limited C uptake rate may be attributed to low light availability or small whole-plant leaf area. In this regard, we introduced

a criterion $PAR_k$, where the photosynthetic rate is reduced by half of the light-saturated photosynthetic rate, allowing the assumption that the limited C uptake rate is due to low light availability if $PAR_{top}$ is lower than $PAR_k$. In this case, the resources are allocated to the stem for height growth to acquire better light conditions under tree competition (Case 5 or 6 in Fig. 3); otherwise, the resources are allocated for an increase in leaf area (Case 3 or 4 in Fig. 3). Lastly, the residual $C_{grow}$ or $N_{grow}$ after the biomass allocation is allocated to the stock pool.

**2.4 Simulation configuration**

The model was applied to the Fukido mangrove forest to test its performance in reproducing the forest structural variables (species composition, mean DBH, and AGB). The model was forced with atmospheric variables (air temperature, relative humidity, atmospheric pressure, wind speed, and cloud fraction) and substrate conditions (soil salinity and porewater DIN). Direct and diffused solar radiation and longwave radiation were calculated in SEIB-DGVM from the given variables such as cloud fraction, air temperature, and latitude (Sato et al., 2007). The atmospheric variables for the Fukido mangrove

forest given to the model were derived from a global reanalysis product JRA-55 (Kobayashi et al., 2015). For long-term simulation (i.e. more than 100 years), the yearly atmospheric variation in 2013, a year when the field-data collection was conducted, was repeatedly given in the simulation.

Simulations with different soil salinity, or the "salinity gradient simulation", which varied from 18‰ to 36‰ with

2‰ intervals, were conducted to reproduce the forest structural variables across a soil salinity gradient. For the porewater DIN, a spatially averaged DIN (average of DIN measured at the survey plots: 200 µmol $L^{-1}$) was given to the model as the representative value of the porewater DIN in this forest. In each simulation, soil salinity and the porewater DIN were set as constant due to lack of data and model on the temporal variations in substrate conditions. We also conducted "plot-wise simulation", or the simulation for each survey plot, by giving the measured soil salinity and porewater DIN. Note that the

results shown in this manuscript are from the "salinity gradient simulation"; the results of the "plot-wise simulation" is provided in Fig. S5 in the Supporting Information and discussed later.

The initial condition was set as bare land (no vegetation) for all simulations. Tree establishment occurs at 1 m × 1 m grid-cells at yearly time step according to light condition at the forest floor and a parameter of establishment probability ($P_{establish}$, $m^{-2}$ $year^{-1}$) prescribed for each species (Sato et al., 2007). The species that will establish at a grid-cell is determined





according to a fraction of total biomass of each species in the computational domain such that a species occupying a larger fraction has a higher probability of establishment. On the other hand, it is sometimes randomly determined by a probability $Est_{random}$, where the value of $Est_{random}$ was set to 0.05 in this study. This corresponds to Scenario 4 in the tree establishment scheme in SEIB-DGVM (see Sato, 2015 for the details). We followed Sato et al. (2007) for the initial conditions (tree morphology and biomass proportion) of the established trees.

The SEIB-DGVM uses stochastic models for the processes of tree establishment and mortality, and for this reason the result of a simulation varies every time. In this regard, we conducted ensemble simulations (20 runs) for each soil salinity in the "salinity gradient simulation" and extracted the general trends.

        The model parameter settings related to plant hydraulics and productivity are summarized in Table 2. Other minor model parameters are summarized in Table S2. The parameter values for the two-species in the Fukido mangrove forest, *R.*

*stylosa* and *B. gymnorrhiza*, were determined based on literatures. If the data for a focal species was unavailable from the literature, the data from the genus or family was applied. Some parameter values were adapted from other mangrove genus or terrestrial ecosystems, and in this case, the same value was given to the two species (Table 2). The values of $\Psi_{lk}$ (critical leaf water potential) and $\beta_0$ (sensitivity of marginal WUE to leaf water potential in Eq. (S21), see Note S3) of each species were calibrated to reproduce the AGB and mean DBH of each species across the soil salinity gradient.

Fukido mangrove forest's age is unknown, which makes the comparison between the model and field-data difficult. However, considering that it is an old and mature forest intact at least since 1977 (Ohtsuka et al., 2019), we assumed that the forest structural variables of the Fukido mangrove forest are in steady states. We conducted long-term simulations for 450 years with this assumption, and extracted the modeled DBH and AGB in steady states (> 300 years) and compared them with the field data.

Table 2. Model parameters related to plant hydraulics and productivity.

| Symbol | Description | Units | R. s | B. g | Source |
|---|---|---|---|---|---|
| $\rho$ | Wood density | g cm$^{-3}$ | 0.84 | 0.76 | Zanne et al. (2009) |
| $SLA$ | Specific leaf area | cm$^2$ g$^{-1}$ | 45 | 71 | Sharma et al. (2012) |
| $\varepsilon$ | Salt filtration efficiency | Fraction | 0.90[a] | 0.99 | Reef and Lovelock (2015) |
| $R_r$ | Fine root hydraulic resistance | MPa s g kg$^{-1}$ H$_2$O | 2220[b] | 2220[b] | Bonan et al. (2014) |
| $K_{sap}$ | Stem hydraulic conductivity | kg H$_2$O m m$^{-2}$ sapwood s$^{-1}$ MPa$^{-1}$ | 1.44[a] | 1.13 | Melcher et al. (2004), Jiang et al. (2017) |
| $P_{50}$ | Water potential at which 50% of xylem conductivity is lost | MPa | -4.4[a] | -8.2 | Melcher et al. (2004), Jiang et al. (2017) |
| $a_2$ | Empirical parameter shaping xylem vulnerability | – | 4.5[a] | 4.6 | Melcher et al. (2004), Jiang et al. (2017) |
| $C_p$ | Plant capacitance | kg H$_2$O m$^{-2}$ leaf MPa$^{-1}$ | 0.045[b] | 0.045[b] | Bonan et al. (2014) |
| $\Psi_{l,min}$ | Minimum leaf water potential | MPa | -4.5[a,c] | -4.0[c] | Hao et al. (2009), Lovelock et |





| | | | | | |
|---|---|---|---|---|---|
| | | | | | al. (2006), Deshar et al. (2008) |
| $\Psi_{lk}$ | Critical leaf water potential | MPa | -3.9 | -3.4 | Calibrated |
| $V_{cmax,25}$ | Maximum carboxylation rate at 25 ℃ | $\mu mol\ m^{-2}\ s^{-1}$ | 50[d] | 50 | Estimated from Ball et al. (1988) |
| $\lambda_0$ | Reference marginal water use efficiency in Eq. (S21) | $\mu mol\ CO_2\ mol^{-1}\ H_2O$ | 250 | 250 | Assumed |
| $\beta_0$ | Sensitivity of marginal water use efficiency to leaf water potential in Eq. (S21) | $MPa^{-1}$ | -0.4 | -0.6 | Calibrated |
| $TO_l$ | Leaf turnover rate | $day^{-1}$ | 0.0024 | 0.0019 | Sharma et al. (2012) |
| $TO_{cr}$ | Coarse root turnover rate | $day^{-1}$ | 0.0003[e] | 0.0003[e] | Castañeda-Moya et al. (2011) |
| $TO_{fr}$ | Fine root turnover rate | $day^{-1}$ | 0.001[e] | 0.001[e] | Castañeda-Moya et al. (2011) |
| NRE | Nitrogen resorption efficiency | fraction | 0.85 | 0.85[f] | Lin et al. (2010) |

a. Value for *Rhizophora magle*

b. The value used for terrestrial forest ecosystem was applied due to lack of information.

c. The minimum of the reported values was adopted.

d. Value for *Rhizophora apiculata*

e. The average value of data in Castañeda-Moya et al. (2011) was adopted.

f. Value for *Rhizophora stylosa*

## 3 Results

### 3.1 Modeled seasonal dynamics

Seasonal variations in atmospheric forcing variables and modeled gross photosynthetic rate ($P_g$) and transpiration (T) normalized by the leaf area index (LAI) are shown in Fig. 4. The modeled variables were from one of the ensemble simulations with soil salinity set as 30‰. The model demonstrated strong seasonality in photosynthesis and transpiration primarily due to seasonality in solar radiation and air temperature. The model predicted the peak of $P_g$/LAI in June with a value of ~ 5.0 g C $m^{-2}$ $day^{-1}$ and the peak in T/LAI in July–Sep with a value ~ 0.9 mm $day^{-1}$. The $P_g$/LAI and T/LAI were predicted to be depressed during winter (December–February) with values ~ 3.0 g C $m^{-2}$ $day^{-1}$ and ~ 0.4 mm $day^{-1}$, respectively. We compared the modeled leaf-level $P_g$ with the field-estimated values in the Fukido mangrove forest by Okimoto et al. (2007). Their measurements were conducted in an area where the LAI is 1.55, the same LAI as the one shown in Fig. 4d; thus, the effects of LAI on leaf-level $P_g$ could be eliminated for comparison. Although the modeled $P_g$/LAI is slightly lower than the one obtained by Okimoto et al. (2007) (~ 1.0 g C m-2 $day^{-1}$), especially from June to August, overall, the model agreed well with their results.
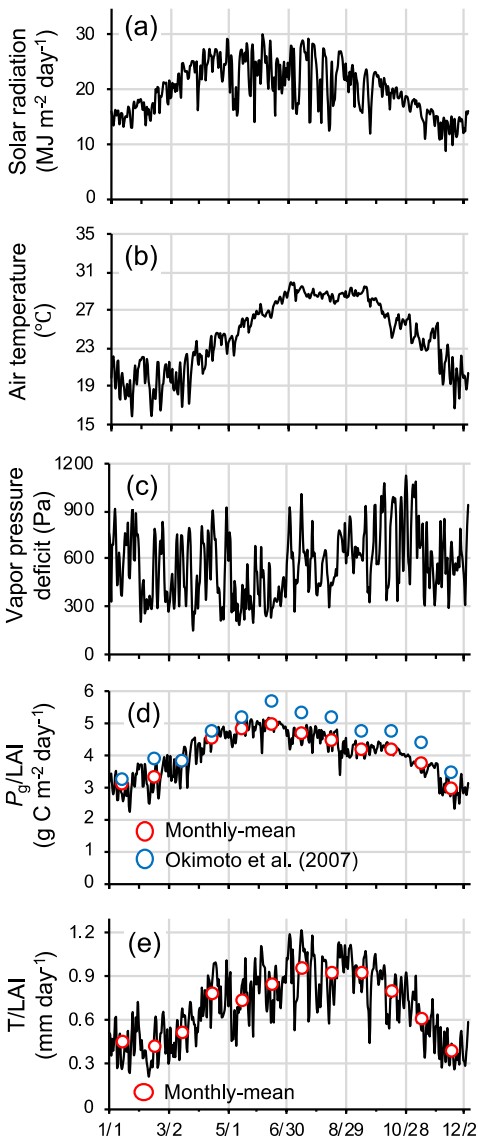

Figure 4. Seasonal variations in atmospheric forcing variables: (a) solar radiation, (b) air temperature, and (c) vapor pressure deficit (VPD), and modeled seasonal dynamics: (d) gross photosynthetic rate ($P_g$, g C m$^{-2}$ ground day$^{-1}$) and (e) transpiration (T, mm day$^{-1}$) normalized by leaf layer index (LAI, m$^2$ leaf m$^{-2}$ ground). Solar radiation is expressed as daily sum while air temperature and VPD are expressed as daily mean. Here, the modeled dynamics were from a simulation of soil salinity set as 30‰, and the results of a year when LAI reached 1.55 were shown. At this time, the LAI of *R. stylosa* and *B. gymnorrhiza* were 0.87 and 0.68, respectively. In panel (d), seasonal variations in GPP/LAI measured by Okiomoto et al. (2007) are also shown as reference, the data of which are from an area with LAI = 1.55 in Fukido mangrove forest in 2000–2001.




## 3.2 Modeled biomass dynamics under different soil salinity

Figure 5 shows the changes in the forest structures for over 200 years under different soil salinity conditions, 20‰,
24‰, 30‰, and 34‰, from one of the ensemble simulations. The time-series results of AGB, LAI, and mean DBH of the
two species are shown in Fig. 6. Trees with DBH < 0.05 m were not accounted for in the calculation of the mean DBH
because it is sensitive to the presence of small trees. Overall, the model demonstrated the significant influence of soil salinity
on species composition and forest structure.

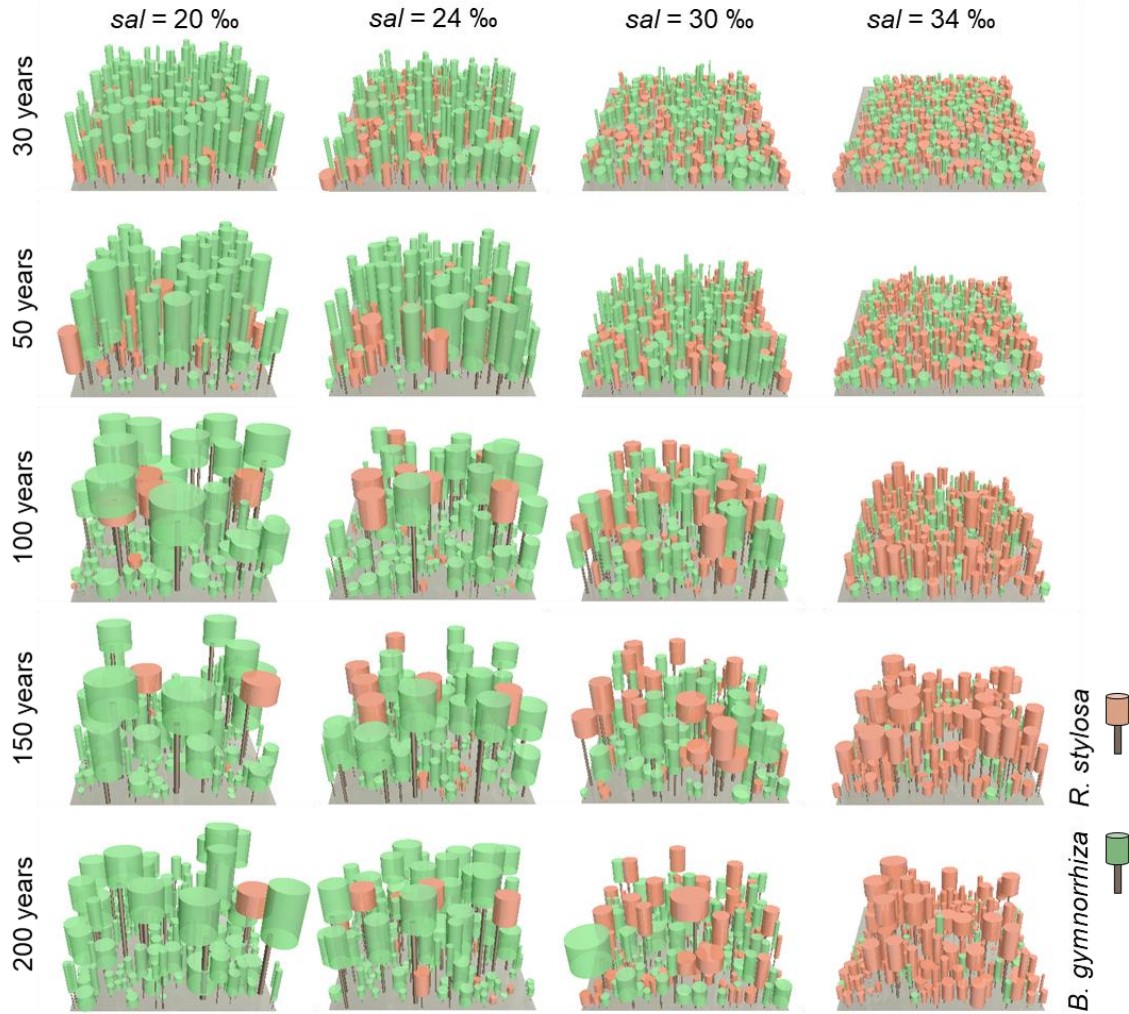

Figure 5. Visualization of forest structures over 200 years under different soil salinity (*sal*), 20, 24, 30, and 34‰, taken from
one of the ensemble simulations. The brown-colored objects represent the stem the while the orange- or green-colored
objects represent the crowns of *R. stylosa* and *B. gymnorrhiza*, respectively. The forest floor shown is the 30 m × 30 m-wide
computational domain.



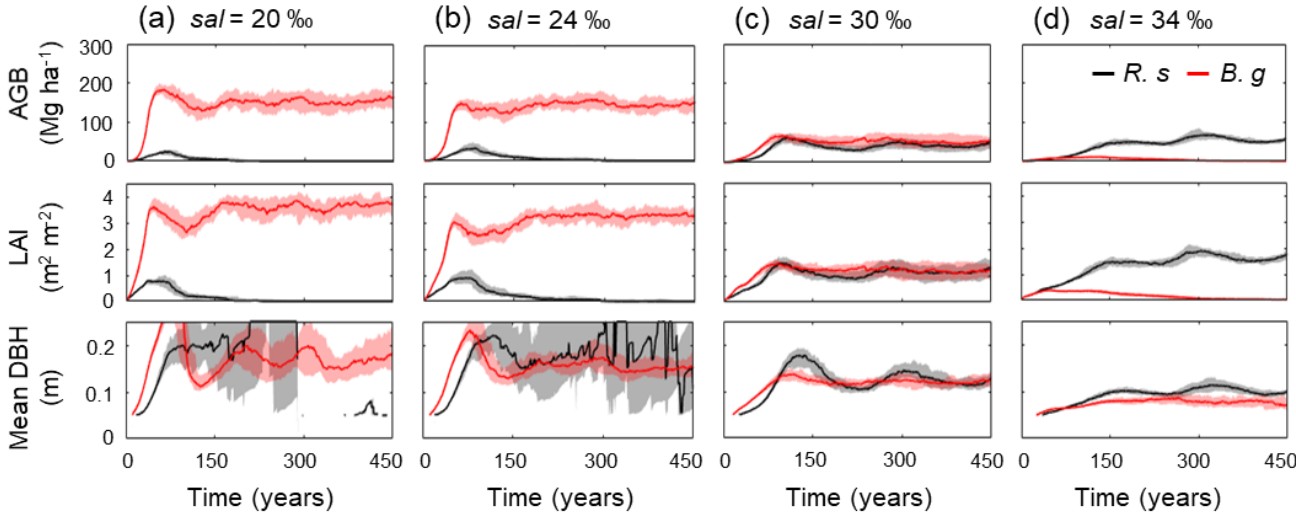

Figure 6. Temporal dynamics in above-ground biomass (AGB), leaf area index (LAI), and mean diameter at breast height (DBH) of *R. stylosa* (*R. s*) and *B. gymnorrhiza* (*B. g*) in four soil salinity conditions (a) 20‰, (b) 24‰, (c) 30‰, and (d) 34‰. Note that trees with DBH < 0.05 m were not included in the calculation of mean DBH. Solid lines show median and shading the 90-th percentile from ensemble simulations.

The model predicted that *B. gymnorrhiza* dominates over *R. stylosa* when soil salinity is 20‰ or 24‰ (Fig. 6a–b).
Under soil salinity of 20‰, the AGB of *B. gymnorrhiza* exponentially increased up to 200 Mg ha$^{-1}$ after 60 years since the initial condition. It slightly decreased after that, and was kept almost constant at 175 Mg ha$^{-1}$ after 150 years. The LAI of this species showed almost the same trend with AGB while the mean DBH showed fluctuation especially in the first 200 years (Fig. 6a). The sudden decrease in the mean DBH is attributed to the onset of deaths of large *B. gymnorrhiza* trees that generated forest gaps and promoted the establishment of small trees (Fig. 5). After the decrease in the mean DBH, it
gradually increased again and saturated at 0.17 m (Fig. 6a). Alternatively, the AGB and LAI of *R. stylosa* were significantly lower than *B. gymnorrhiza* with its peak at only 25 Mg ha$^{-1}$ and 1 m$^2$ m$^{-2}$, respectively. This can also be seen in the decreasing number of *R. stylosa* trees subsequent to forest growth (Fig. 5). In contrast to AGB and LAI, the mean DBH of *R. stylosa* reached around 0.2 m after 75 years, as large as that of *B. gymnorrhiza* in steady state. This suggest that some *R. stylosa* trees can grow until mature conditions (see also Fig. 5), while trees of this species with DBH > 0.05 m disappeared in
all ensemble simulations after 300 years (Fig. 6a). The trees of *R. stylosa* sometimes emerge due to the random factor in the establishment process, but most of the trees did not grow more than DBH of 0.05 m in the canopy of *B. gymnorrhiza*.

The trend in forest growth under 24‰ salinity was similar to that of 20‰ (Fig. 5, Fig. 6b), but showed a slightly lower and higher peak for *B. gymnorrhiza* and *R. stylosa*, respectively, of the AGB, LAI, and mean DBH. This suggests decreased productivity of *B. gymnorrhiza* compared to soil salinity 20‰, and increased productivity of *R. stylosa* albeit the





increase in soil salinity. The survival rate of *R. stylosa* was higher than the results for 20‰, resulting in the high mean DBH of this species throughout the simulation period (Fig. 6b).

When the soil salinity was 30‰, the AGB of *B. gymnorrhiza* significantly decreased compared to the results for salinities 20‰ and 24‰, becoming equivalent to those of *R. stylosa* (Fig. 6c). The LAI and mean DBH also showed a significant decrease, suggesting significantly lowered productivity of *B. gymnorrhiza*. The AGB and LAI of *R. stylosa*

significantly increased compared to the results for 20‰ and 24‰, but the mean DBH significantly decreased.

The model predicted that *B. gymnorrhiza* cannot grow well at soil salinity 34‰, and that *R. stylosa* dominates under this salinity condition (Fig. 5, Fig. 6d). Despite the further decrease in AGB, LAI, and mean DBH of *B. gymnorrhiza*, those of *R. stylosa* showed almost the same level for these parameters at soil salinity 30‰.

### 3.3 Comparison between modelled and field-measured forest structural variables

Figure 7 shows the field-measured and modeled mean DBH and AGB of *R. stylosa* and *B. gymnorrhiza* across the soil salinity gradient. The field data clearly showed the effects of soil salinity on forest structural variables – decrease in mean DBH for both species, and decrease in AGB of *B. gymnorrhiza* but increase in AGB of *R. stylosa* with increasing soil salinity. The model reproduced well the said patterns across the soil salinity gradient and the values are within or close to the field-data variations. The change in species composition is also well-reproduced, suggesting that the model can reproduce

the forest structural variables across the soil salinity gradient.

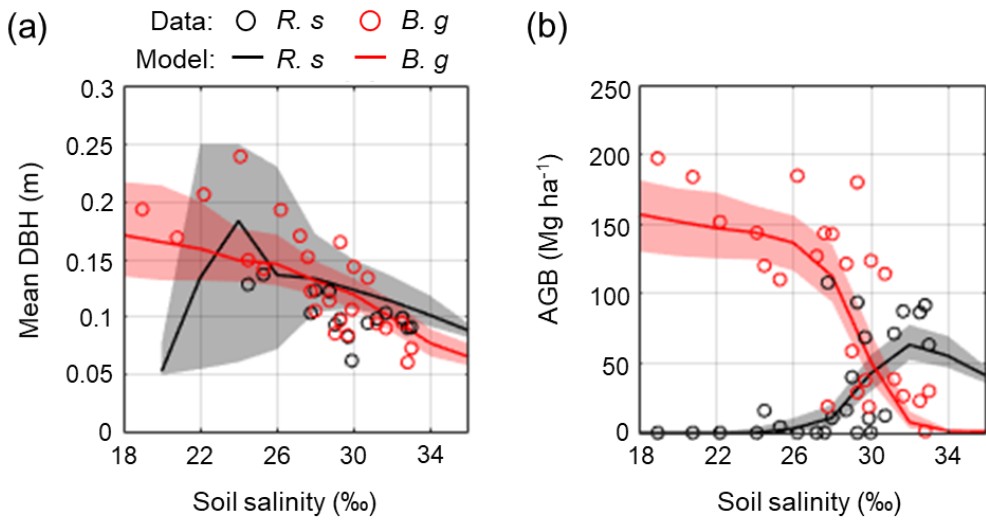

Figure 7. Comparison of field-measured and modeled (a) mean DBH and (b) AGB of *R. stylosa* and *B. gymnorrhiza* along with soil salinity gradient. From each ensemble simulation, modeled mean DBH and AGB in steady states (> 300 years) were extracted and pooled for all ensembles, and the median (solid line) and the 90-th percentile (shading) of the pooled

samples were shown. Note that trees with DBH < 0.05 m were not included in the calculation of the mean DBH.

Figure 8 shows the field-measured and modeled relationship of tree density and mean individual stem biomass. Although there are some discrepancies between the model and field data especially for conditions soil salinity > 30‰, the model reproduced the overall pattern of the field data.

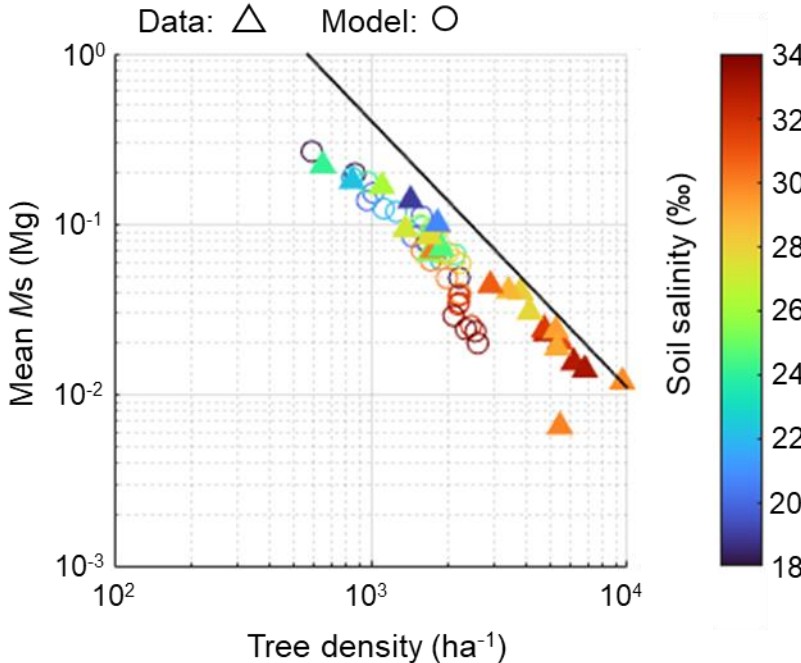

Figure 8. The relationship of tree density and mean individual stem biomass ($M_s$). Triangles show field data while circles show modeled values from one of the ensemble simulations with different soil salinity settings (from 18‰ to 34‰ with 2‰ increments) plotted from 300–450 years (with interval of 50 years), which is in steady states in terms of forest structural variables (see Fig. 6). Note that trees with DBH < 0.05 m were not counted in calculating tree density and mean $M_s$. The line represents the full density curve proposed by Tabuchi et al. (2013): $y = 20389x^{-1.567}$.

**3.4 Modeled morphological traits and effects of soil salinity**

We examined the modeled tree morphology such as shoot/root biomass (S/R) ratio, leaf area/sapwood area (*LA/SA*) ratio, and tree height (Fig. 9). The model predicted the general trends of increasing S/R ratio, decreasing *LA/SA* ratio, and increasing *H* following the growth of trees. These results suggest biomass allocation patterns such as an increase in biomass allocation to the stem relative to the roots and leaves (Fig. 9a–d). The modeled DBH-*H* agreed with the field data-derived general DBH-*H* relationship while the results for *R. stylosa* appeared overestimated compared to the data (Fig. 9e–d). Fluctuations, especially in the DBH-*H* relationship of *B. gymnorrhiza* under soil salinity 34‰, are attributed to the small number of trees, which may have been insufficient to capture the trend in tree morphology.



The modeled tree morphology showed significant variations among different soil salinities, especially *B. gymnorrhiza* (Fig. 9). The model predicted a decrease in S/R ratio, *LA*/*SA* ratio, and *H* relative to DBH with an increase in

soil salinity.

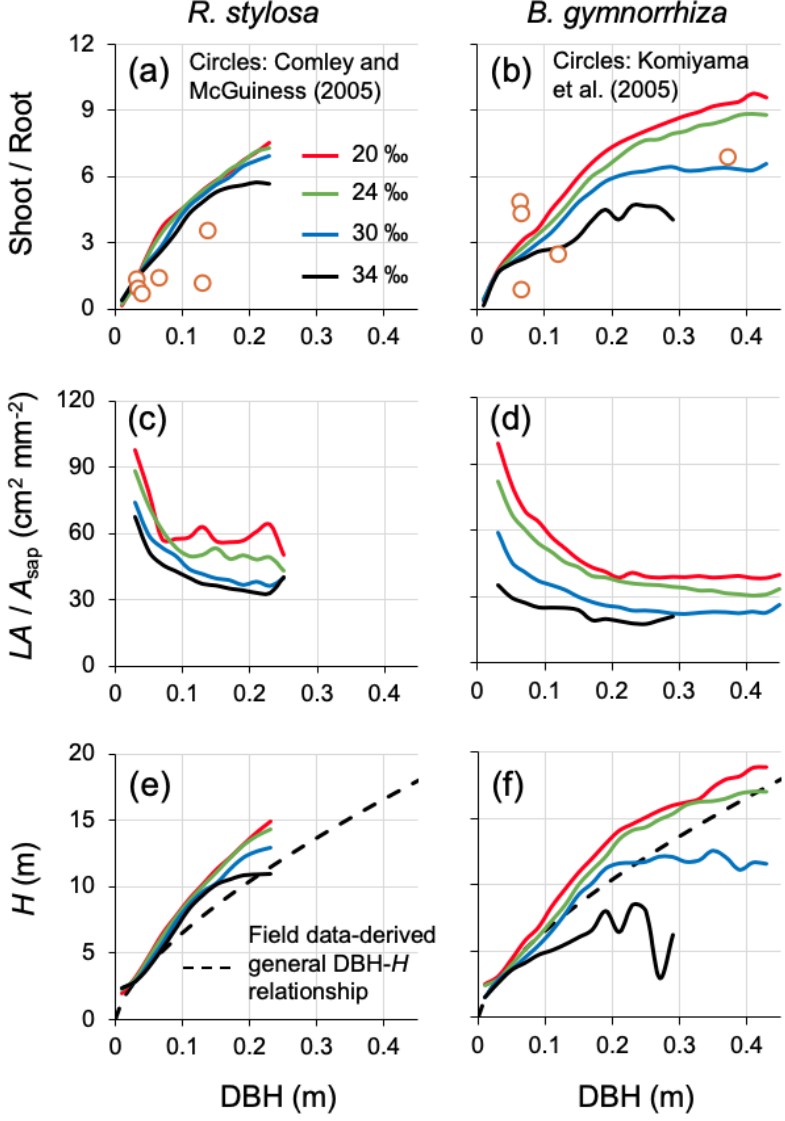

Figure 9. Modeled relationships of (a, b) shoot/root biomass ratio, (c, d) whole-plant leaf area (*LA*)/sapwood (*A*$_{sap}$) ratio, and (e, f) tree height (*H*) with DBH under different soil salinity conditions. From each ensemble simulation result from 300–450 years, which is in steady states in terms of forest structural variables, the modeled individual trees' variables were extracted

every 10 years. The extracted samples were pooled for all ensemble simulations. The pooled samples were then binned with DBH width of 0.02 m, and the median value in each bin was shown. Here, the shoot biomass refers to the sum of stem and





leaf biomass, and the root biomass refers to the sum of coarse and fine root biomass. Note that the above-ground root biomass (of *R. stylosa*) is not included in the shoot biomass both in the model result and the data of Comley and McGuiness (2005). Also, note that Komiyama et al. (2005) data include *B. gymnorrhiza* and *B. sexangla*. See Note S1 and Fig. S1 in the

Supporting Information for field-data derived general DBH-*H* relationship details.

## 4 Discussion

### 4.1 Model performance

Forest growth is influenced by leaf-level and whole-plant $CO_2$, water and nutrient fluxes, and forest-scale tree competition, which are all interconnected. The leaf-level fluxes were simulated using a well-established stomatal

optimization scheme with the marginal WUE linked with leaf water potential (Bonan et al., 2014; Xu et al., 2016). The model predicted the distinct seasonal dynamics in photosynthesis and transpiration in the Fukido mangrove forest (Fig. 4). The modeled seasonal variations in leaf-level photosynthesis ($P_g$/LAI) agreed well with the one measured by Okimoto et al. (2007) in this forest (Fig. 4d). Although there are no data on the seasonal variations in transpiration in this forest, studies on other subtropical mangrove forest, such as the Everglades National Park, Florida (Barr et al., 2014), and China (Liang et al.,

2019) that incorporated the eddy-covariance approach also showed strong seasonality in transpiration, similar to the one predicted for the Fukido mangrove forest in this study (Fig. 4e). The evapotranspiration rate normalized by LAI in the Everglades measured by Barr et al. (2014) was 0.4–1.2 mm day$^{-1}$, which is close to the variation of the modeled T/LAI in the Fukido mangrove forest (Fig. 4e). These results suggest that the model produced realistic seasonal dynamics for transpiration in the Fukido mangrove forest.

Tree growth was driven by C and N uptake rates in the developed model resulting from the leaf-level and the whole-plant $CO_2$ and water fluxes. The modeled growth rates at a soil salinity condition where *B. gymnorrhiza* is the dominant species (*sal* < 28‰) showed close values to the ones measured by Ohtsuka et al. (2019) at a *B. gymnorrhiza*-dominated site in the Fukido mangrove forest (Fig. S4). This suggests that the model also reasonably predicted the growth rate of each species in addition to the leaf-level $CO_2$ and water fluxes.

This model also showed reasonable reproducibility of the tree density-mean $M_s$ relationship, except for those with soil salinity > 30‰. An exponent value close to -3/2 was obtained, similar to what is observed in the Fukido mangrove forest (Suwa et al., 2021; Fig. 8) and in many mangrove forests as well (Analuddin et al., 2009; Deshar et al., 2012; Khan et al., 2013; Tabuchi et al., 2013; Azman et al. 2021). This was achieved by implementing the species-specific morphological traits especially the DBH-$D^*_{crown}$ relationship (Fig. 3b, see also Note S1 and Fig. S1), suggesting that the self-thinning process

arising from the tree competition was simulated well by the model. The deviation of the modeled tree density-mean $M_s$ relationship from the data at high soil salinity condition may be attributed to the inaccurate representation of the crown morphological trait of *R. stylosa* (see Note S1 and Fig Sc). Crown size representation could be a factor that drives a large part of the uncertainty in DVMs (Meunier et al., 2021). Nevertheless, the data are remarkably scarce in the case of





mangroves. The morphological traits of crown size should be investigated in future studies for more realistic representation
of mangroves' tree competition in the model.

        Overall, this is first modeling study to introduce detailed physiological and mechanistic representations of the
mangrove forest growth controlled by photosynthesis, water and nutrient (represented by DIN) uptake, tree competition, and
achieved good as well as comprehensive reproducibility of mangrove growth processes. The remarkable agreement of
modeled forest structures with field data across a soil salinity gradient validated our hypothesis – individual-based DVM
incorporating plant hydraulic traits can reasonably predict mangrove growth processes under salt stress without empirical
expression of the soil salinity influence on mangrove productivity. However, the model still does not account for the plant-
to-soil feedback through water uptake, which has been identified by a mangrove growth-groundwater flow coupled model
(Bathmann et al., 2021) as an important factor affecting both mangrove and substrate conditions (soil salinity). Alternatively,
the said model also demonstrated that the forest structural variable and soil salinity dynamics could reach steady states after
some time from the initial condition, a setting that is considered to describe the Fukido mangrove forest (Ohtsuka et al.,
2019). Our modeling results, which did not include the plant-to-soil feedback, therefore may be valid only for the steady
states and still holds uncertainty in the developmental stage. This further implies that model application may be limited only
to mature mangrove forests, and further model improvement is needed for its application to forests during the developmental
stage (after plantation) or during the recovery stage (after disturbances such as typhoons and deforestation).

**4.2 Soil salinity and interspecific competition shaping the forest structural variables**

        Overall, the model explained that the changes in mean DBH and AGB of the two coexisting species with change in
soil salinity are due to the difference in their salt tolerance and interspecific competition (Figs. 6, 7). Specifically, the model
predicted that *B. gymnorrhiza* competes over *R. stylosa* when soil salinity is favorably low for the growth of *B. gymnorrhiza*
(*sal* < 28‰), an observation that is consistent with our field data and the data from other mangrove forests (Putz and Chan,
1986; Enoki et al., 2014). This result may be attributed to the following model parameter settings based on literature – higher
wood density ($\rho$), smaller specific leaf area (SLA), and higher leaf turnover rate ($TO_l$) of *R. stylosa* than *B. gymnorrhiza*
(Table 2). Higher $\rho$ indicates the requirement of higher biomass increase for the height or radial growth of the stem. Smaller
SLA and higher $TO_l$ indicate the higher requirement of C and N to produce new leaf tissues or to keep the same amount of
leaves, i.e. the need of *R. stylosa* for more C and N resources for growth compared to *B. gymnorrhiza*. The biomass
requirement of prop roots, which lowers the biomass allocation to the stem (Fig. S3), and the smaller $D^*_{crown}$ of *R. stylosa*
compared to *B. gymnorrhiza* (Fig. S1c–d) may also have contributed to the former's lower growth rate. Consequently, *B.
gymnorrhiza* grew faster and suppressed the growth of *R. stylosa* by severe shading (Fig. 5, 6). The higher growth rate of *B.
gymnorrhiza* compared to *R. stylosa* at relatively low salinity conditions agrees with the study by Jiang et al. (2019).

        Interestingly, our model was able to simulate unique conditions not previously reported by other modeling works.
For instance, the model predicted that *R. stylosa* trees could grow until the mature conditions under the canopy of *B.
gymnorrhiza*-dominated forest provided the chance of favorable light conditions, resulting in the high mean DBH but low





AGB of this species at relatively low soil salinity (∼ 24‰) (Fig. 5, 6). Simulating this kind of process may only be possible through the individual-based approach with calculations of detailed irradiance distribution as done by the SEIB-DGVM in this study. Alternatively, the model predicted the significantly lowered growth rate of *B. gymnorrhiza* at high soil salinity

condition (*sal* > 30‰) where *B. gymnorrhiza* cannot grow until mature conditions, which resulted in the low AGB and small mean DBH of this species. This reduced the suppression of *B. gymnorrhiza* on *R. stylosa* and generated the *Rhizophora stylosa*-dominated forest (Fig. 5, 6). Despite the abundant population of *R. stylosa*, the sizes of individuals were relatively small due to high salt stress, and resulted in the high AGB but small mean DBH of this species.

### 4.3 Implications of the predicted morphological traits

We introduced a novel biomass allocation scheme that adjusts plant hydraulics and enhances the uptake rate of growth-limiting resources (C or N). The biomass allocation is determined on the basis of a few thresholds ($\Psi_{lk}$, $PAR_k$; Fig. 3), and does not need prescribed allometry on leaf and root biomass unlike some DVMs (e.g., ED2 by Medvigy et al. 2009, the original SEIB-DGVM by Sato et al., 2007), the data of which are usually limited and labor-intensive to obtain in the field. This approach also holds potential to describe the plants' morphological adaptation to changing environments (Poorter

et al., 2012). While previous studies have already linked plant hydraulics to biomass allocation and showed high determination of the allocation pattern (Magnai et al., 2000; Trugman et al., 2019b; Portkay et al., 2021), the model presented has novelty in its consideration of N-limited growth and different stoichiometry (C/N ratio) among plant organs, which influences hydraulics optimization (indicated by Eq. (8)). Additionally, this study introduced the flexibility in the optimization of the DBH-*H* relationship such that plants adjust their hydraulics not only from the root biomass but also from

the sapwood area (Fig. 3).

The forest-scale S/R ratio of mangroves is usually within the range of 1–10, and sometimes shows values greater than 20 or less than 0.5 (Comier et al., 2015; Adame et al., 2017). Predicted values for individuals are within these reported values, suggesting that the model constrained the biomass allocation process well (Fig. 9a–b). Interestingly, the model predicted an increase in biomass allocation to the stem relative to the roots as mangroves grow, resulting in increasing S/R

ratio with an increase in DBH, an observation that is consistent with the general trend of plant biomass allocation (Poorter et al. 2012; Portkay et al., 2021). Although the model seemed to have somewhat overestimated the S/R ratio of *R. stylosa*, the overall agreement with the trend of the data of Comley et al. (2005) and Komiyama et al. (2005) suggests that the biomass allocation scheme presented in this study successfully captured the allometric relationship in biomass proportion.

Validation of the predicted S/R ratio, aside from it being beyond the scope of this manuscript, cannot be done due to

lack of available data; however, the model provided some implications on morphological adaptations to soil salinity. This can be seen in the increase in root biomass relative to shoot biomass with an increase in soil salinity (Fig. 9a–b), suggesting the effectiveness of root allocation for the optimization of plant hydraulics under enhanced salt stress. Similarly, the adaptation can be seen in decrease in *LA* /sapwood area ratio (Fig. 9c–d) and tree height (Fig. 9e–f) relative to DBH with an increase in soil salinity, which in turn decreased the whole-plant transpiration demand and increased the hydraulic





conductivity. Several studies have reported morphological plasticity of mangroves in relation to soil salinity, such as changes in S/R ratio (Ball et al., 1997; Nguyen et al., 2015; Chatting et al., 2020) and DBH-$H$ relationship (Suwa et al., 2008, 2009; Vovides et al., 2014; also see Fig. S1a–b) that support the model implications. However, more studies are needed for deeper understanding and model improvement of the mangrove biomass allocation dynamics in relation to size and environmental conditions (e.g., nutrient availability). Specifically, the effects of varying size and environment settings should be considered

separately in future studies because the biomass proportion is basically a function of size (Fig. 9a–b).

## 4.4 Effects of other factors and further model improvement

    Besides soil salinity, this study highlighted the importance of atmospheric variables as important drivers controlling mangrove production. This is seen in the photosynthesis-transpiration seasonal dynamics with peak during summer (June–September) and depression during winter (November–March) (Fig. 4) that none of the previous mangrove modeling studies

has examined yet. The model predicted winter depression primarily due to low solar radiation and air temperature. Specifically, low air temperature (< 20 ℃) significantly reduced photosynthetic capacity – the maximum carboxylation rate and the maximum electron transport rate (Aspinwall et al., 2021); this, in turn, decreased the marginal WUE ($\Delta A_n/\Delta E$), leading to the downregulation of stomatal conductance, a behavior of mangroves' stomata observed under low temperature conditions (Akaji et al., 2019; Aspinwall et al., 2021). This resulted in the depression of photosynthesis and transpiration

during this season. Such winter depression lowers the production of mangroves in subtropical regions, and may be differentiated from tropical mangroves in terms of productivity. This could be a key factor in explaining and predicting the latitudinal gradients in mangroves' structural variables such as canopy height and AGB with the highest values at the equatorial region (Saenger and Snedaker, 1993; Simard et al., 2019; Rovai et al., 2021).

    The role of N uptake in mangrove growth was explicitly modeled in this study for the first time. The model

predicted that the annual mean of ecosystem-level net C and N uptake ratio in the Fukido mangrove forest at soil salinity 30‰ is around 230, indicating N-limitation for leaf and root tissue development (Table 1). This suggests the importance for N in predicting mangrove growth in this forest, as in many other mangrove forests (Reef et al., 2010).

    The model gave significantly better prediction of the AGB spatial distribution when the spatially averaged DIN concentrations were applied to the substrate condition compared to plot-wise DIN values (Fig. S5, "plot-wise simulation").

This suggests that N availability was better represented by the spatially averaged value in this study. Porewater DIN in mangrove forests is highly heterogeneous horizontally (Inoue et al., 2011) and vertically (Kristensen et al., 1998; Lee et al., 2008) even in very small scales such as 10 cm. The DIN measured from one soil core sample might not have captured the representative value at each plot due to such heterogeneity. Differences in the predicted AGB between the two cases highlight nutrient availability in affecting mangrove production and biomass dynamics in this forest. Therefore, an

appropriate representation of nutrient availability is critical for accurate prediction of mangrove production. More detailed measurement of porewater nutrient concentrations in space and time is needed for a more reliable model prediction, and future works will account for this aspect. Similarly, future works should consider biogeochemical processes which control

nutrient dynamics in the substrate. For example, the porewater of the Fukido mangrove forest is rich in ammonia compared to nitrate (Table S1), contrary to the groundwater flowing into this forest, which is rich in nitrate (Mori et al., unpublished

data). This suggests that biogeochemical processes, such as mineralization of organic matter, N fixation, and denitrification (Reef et al., 2010) are important drivers controlling nutrient dynamics in the forest, which ultimately affects soil organic matter dynamics. These factors should therefore be taken into consideration in future works as one of the plant-to-soil feedbacks in addition to water uptake processes.

## 5 Concluding remarks

565         This manuscript presents a new individual-based model modified from SEIB-DGVM for a better physiological representation of mangrove growth under the impact of soil salinity. The plant hydraulics was incorporated and linked with the plant production process (C and N uptake) and biomass allocation. The developed model showed high reproducibility of the complex nonlinear patterns in species composition and forest structural variables in a subtropical mangrove forest shaped across a soil salinity gradient without empirical parameterizations of soil salinity influence on mangrove productivity. While

there are still some important processes to be accounted for to further improve the model (e.g., plant-to-soil feedback and soil biogeochemical processes), the physiologically-improved model predicted the various key ecological processes – seasonal dynamics in photosynthesis and transpiration, interspecific competition, self-thinning process, biomass allocation pattern, and morphological adaptation to soil salinity – together with forest structure. Although the model has been tested using only two species in one site, owing to its physiological principles which do not hold empirical expressions of

influences of environmental variables on mangrove productivity, it can be potentially extended to other mangrove species in various environmental settings. Therefore, it may contribute in predicting how the mangrove biomass dynamics will respond to future changes in the global climate.

## Code Availability

Code used in this study can be made available upon reasonable request to the corresponding author.

## Data Availability

The tree census data used in this study are published in Suwa et al. (2021). The tree crown data in Ishigaki Island and model output data can be made available upon reasonable request to the corresponding author.

## Author contributions

MY, TN, SS and KN designed the study. SS and JY conducted field works. All authors contributed to model development,

and MY performed the analyses. TN, RS and KN contributed to result interpretation. MY wrote the manuscript and SS contributed to reviewing and editing.



**Competing interests**

The authors declare that they have no conflict of interest.

**Acknowledgments**

We are grateful to the Japan International Cooperation Agency (JICA) and Japan Science and Technology Agency (JST) through the Science and Technology Research Partnership for Sustainable Development Program (SATREPS) for financially supporting the Project "Comprehensive Assessment and Conservation of Blue Carbon Ecosystems and their Services in the Coral Triangle (*Blue*CARES)". This work was also supported by JSPS KAKENHI Grant Numbers JP25257305, JP12F02371, JP15H02268, and JP20K14835. We thank R. Basina and Dr. Y.H. Primavera-Tirol for their help in collecting
crown data in Bakhawan Ecopark, and Y. Marini for providing the crown data in Karimunjawa Island. We also thank Dr. C. Ferrera for providing language help.

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
