# Peer review of "Predicting mangrove forest dynamics across a soil salinity gradient using an individual-based vegetation model linked with plant hydraulics"

_Biogeosciences, 2021_

## Author Comment (AC1)

Response to community comment (CC) posted by Dr. Md Nabiul Islam Khan

We would like to thank Dr. Khan for his interest in our manuscript and for his valuable comments. Please see our answers to the provided comments below.

CC: Figure 7. Comparison of field-measured and modeled (a) mean DBH and (b) AGB of *R. stylosa* and *B. gymnorrhiza* along with soil salinity gradient. How to confirm that the observed pattern is ONLY due to salinity? What about other driving force to make this pattern?

Response:

We do not intend to argue that the observed patterns in the forest structures were shaped only by the salinity gradient. Actually, the observed AGB and mean DBH showed some variations even at the same salinity levels, and such variations may have been due to other abiotic and biotic factors.

We thought that the spatial variations in nutrient availability (porewater dissolved inorganic nitrogen concentration, DIN) may explain such variations. However, as shown in the Fig. S5 of the "plot-wise simulation", the spatial variations in DIN did not improve the predictions of the forest structures. Please see Ln. 548–557 for the discussion about this point.

There are some other possible factors that may have affected forest structures (e.g., hydroperiod, historical disturbances, forest ages, phosphorus availability), but we cannot assess those effects because of lack of information and knowledge for the site. These effects would be examined in future studies.

Nevertheless, salinity is clearly the major factor that have shaped the forest structural patterns as shown in Fig. 7, and the model predicted well the general patterns observed across the salinity gradient, which is the primary scope of this study.

As an action for manuscript revision, we will include sentences to make this point clear.

CC: Figure 6. Temporal dynamics in above-ground biomass (AGB). The scenario (d) shows a low AGB but still showing a reasonable LAI in 100 years simulation. This low AGB doesn't correspond to Figure 5, where vegetation cover indicates a high AGB.

Response:

The results shown in Figs. 5 and 6 are consistent.

First, we assume that this comment is about the model results of *Rhizophora stylosa*.

We think this comment came out because of the axis scaling of Fig. 6, which made it difficult to see the small variations of AGB in short time scale (e.g., 50 years). We provide below the temporal dynamics of AGB, LAI, and mean DBH for 34‰ salinity magnified for the first 200 years (Fig. R1).

This figure shows more clearly the substantial increase in AGB of *R. stylosa* from 50 years to 100 years, which corresponds to the results shown in Fig. 5.

Because Fig. 6 was intended to show long-term dynamics and compare among different salinity conditions, we would like to keep the axis scaling as it is.

About the point "The scenario (d) shows a low AGB but still showing a reasonable LAI in 100 years simulation.", we cannot assess the accuracy of the predicted LAI at this moment because of the lack of observed/field data during study site, which we would like to consider as one of the future tasks through monitoring. Nevertheless, we compared the simulated relationship between AGB and LAI with published data from other mangrove forests (Fig. R2). Although these may not be comparable

due to the different environmental settings and species, the simulation showed reasonable prediction of the general trend of AGB–LAI relationship. Therefore, we consider that the predicted LAI and AGB are reasonable.

We hope this clears the question given for Figs. 5 and 6.

As an action for manuscript revision, we will include the statement that the simulated LAI has not been validated and will be considered for future study.

[Figure]

Figure R1: Temporal dynamics in (a) above-ground biomass (AGB), (b) leaf area index (LAI), and (c) mean diameter at breast height (DBH) of *Rhizophora stylosa* (*R. s*) and *Bruguiera gymnorrhiza* (*B. g*) in soil salinity (*sal*) condition 34 ‰. Trees with DBH < 0.05 m were not included in the calculation of mean DBH. The results are from a simulation corresponding to the one shown in Figure 5.

[Figure]

Figure R2: Simulated yearly trajectory of the relationship between AGB and LAI of *R. stylosa* under the soil salinity 34 ‰ for the first 200 years, which corresponds to the results shown in Fig. R1a-b (red circles). Data from Sharma et al. (2017) (black circles) and Salmo et al. (2013) (triangles) are also shown as reference.

References

Salmo, S. G., Lovelock, C., & Duke, N. C. (2013). Vegetation and soil characteristics as indicators of restoration trajectories in restored mangroves. Hydrobiologia, 720(1), 1-18.

Sharma, S., Nadaoka, K., Nakaoka, M., Uy, W. H., MacKenzie, R. A., Friess, D. A., & Fortes, M. D. (2017). Growth performance and structure of a mangrove afforestation project on a former seagrass bed, Mindanao Island, Philippines. Hydrobiologia, 803(1), 359-371.

---

## Author Comment (AC2)

Response to Referee #1 comment (RC1)

We would like to thank the reviewers for taking the time to review our manuscript and providing constructive comments. Please see our responses to the comments below.

RC1:
Summary
Yoshikai et al. provide a novel modeling study to understand and predict mangrove forest dynamics across a soil salinity gradient. The study added a plant hydraulic module, dynamic allocation module, and nutrient (nitrogen) limitation on growth into SEIB-DGVM. The new model allows for consideration of soil salinity effects on plant ecophysiology as well as soil nutrient levels in mangrove forests. After calibration of two parameters determining allocation and stomata that are unavailable from literature, the model can well represent the spatial gradient of forest structure (mean DBH) and biomass (AGB) across a salinity gradient in a mangrove forest in Japan. Other model-data comparison is also presented. Altogether, the authors conclude that including hydraulic trade-offs and differences in the ability to deal with salinity is critical and adequate for predicting dominant forest dynamics in mangrove forests.

Comments
I really like the study, which extends the existing plant hydrodynamic modeling framework (often used and calibrated in arid/semi-arid ecosystems) to coastal saline ecosystems (also water-stressed). The idea of plant hydraulic control on mangrove forest dynamics existed for some time but the study presents a novel modeling study to evaluate the idea together with field data. Overall, the manuscript is well written and includes adequate details for understanding the model. I have three major comments about model diagnostics, which hopefully can improve the manuscript.

Response:
Thank you very much for the positive assessment and constructive comments on our manuscript. We have addressed the comments suggested by the reviewer, as follows.

RC1:
First, in my opinion, the key evidence to the manuscript's conclusion is Fig. 7&8, which shows how simulated forest structure and biomass match with observed values across the salinity gradient after only modest model tuning (2 parameters in Table 2). However, it is always more important and interesting to know why the model can reproduce the observations. What trait/parameters/processes are dominant in driving the model output. Is it salt filtration efficiency? $P_{50}$?, $\psi_{lk}$?, or $\beta_0$? I would suggest running some sensitivity tests to show what traits/parameters lead to the pattern in Fig.7 and how important is the tuning of $\psi_{lk}$ and $\beta_0$ (their differences seem to be small). In fact, I am curious about whether salt filtration efficiency or $P_{50}$ is more important, or maybe they have to be coordinated in the model to explain the observed pattern. Such information will make the study more useful.

Response:
    We have conducted sensitivity analysis of the plant hydraulic trait parameters ($\varepsilon$, $P_{50}$, $\psi_{lk}$, and $\beta_0$) to see the relative importance of each parameter in reproducing the observed pattern of the forest structure across a soil salinity gradient as suggested. We specifically looked into the sensitivity of

above-ground biomass (AGB), which showed contrasting changes of the two species (*Rhizophora stylosa* and *Bruguiera gymnorrhiza*) with changes in soil salinity in the forest (as shown in Fig. 7b). Please note that to examine the sensitivity of $\psi_{lk}$, we changed the values of both $\psi_{lk}$ and $\psi_{l,min}$ to keep the buffers between the two parameter values; the decrease in $\psi_{lk}$ without decrease in $\psi_{l,min}$ may otherwise lead to the xylem water conductance more susceptible to the water potential at which stomata closes.

For the analysis, we changed the value of a target parameter of one species (either *R. stylosa* or *B. gymnorrhiza*) to the one determined for the other species which is shown in Table 2, and run the "salinity gradient simulation". To save on computational cost, we run only one simulation for each sensitivity test instead of the ensemble approach done for reproducing the forest structures as shown in Fig. 7. The omission of ensemble runs resulted in some fluctuations in AGB along the soil salinity gradient (Fig. R1). However, the fluctuations were not at a level that could affect the interpretation of the overall simulated forest structural patterns across the soil salinity gradient.

The results showed that the change in the values of the parameters $\psi_{lk}$ and $\psi_{l,min}$ had the most impact on the simulation results. The decreases in $\psi_{lk}$ and $\psi_{l,min}$ of *B. gymnorrhiza* to the level determined for *R. stylosa* largely increased the salt tolerance of this species and resulted in the *B. gymnorrhiza*-dominated forest even at the high soil salinity conditions (i.e., > 34 ‰) (Fig. R1f). On the other hand, the increase in these parameters for *R. stylosa* to the level of *B. gymnorrhiza* reduced the salt stress tolerance of *R. stylosa* and resulted in the unsuccessful growth of this species even at the soil salinity higher than 30 ‰ where *R. stylosa* starts to dominate in the forest (Fig. R1e). These results indicate that the mangroves capacity in reducing the leaf water potential is one of the most important functional traits characterizing their salt tolerance as suggested by Reef and Lovelock (2015). The response of AGB to changes in $\psi_{lk}$ also indicates the substantial impact of biomass allocation dynamics determined by $\psi_{lk}$ on plant productivity.

The parameter that have impacted the simulation results next to $\psi_{lk}$ and $\psi_{l,min}$ was the salt filtration efficiency, $\varepsilon$ (Figs. R1a–b). The results shown in Figs. R1a–b highlighted the benefit of partial uptake of salt and associated reduction in xylem tension of *R. stylosa* for maintaining the productivity under the relatively high soil salinity (i.e., > 32 ‰). The changes in the values of $P_{50}$ also affected the simulation results to some extent (Figs. R1c–d). The increase in $P_{50}$ of *B. gymnorrhiza*, which increases the vulnerability to xylem cavitation, decreased the productivity of this species, and resulted in the *R. stylosa*-dominated forest at soil salinity 30 ‰ where the two species showed the same level of AGB in the simulation result shown in Fig. 7. The decrease in $P_{50}$ of *R. stylosa* increased the AGB of this species by around 15 Mg ha$^{-1}$ compared to the case shown in Fig. 7. While the model demonstrated the relatively high sensitivities to these parameters ($\varepsilon$ and $P_{50}$), it is considered that these are the coordinated functional traits, i.e., the lower cavitation resistance (as indicated by higher $P_{50}$ of *R. stylosa*) may result from incomplete salt removal (as indicated by higher $\varepsilon$ of *R. stylosa*) that reduces xylem tension required to maintain water uptake (Jiang et al., 2017). Therefore, they may have to be defined as coordinated plant functional traits resulted from adaptation to salt stress in the model.

The sensitivity of AGB to $\beta_0$ turned out to be quite low suggesting that the choice of -0.6 for $\beta_0$ already leads to efficient stomatal openings for photosynthesis compared to the case of -0.4 for $\beta_0$ (Fig. R1g–h). This may explain the small variations in the leaf-level photosynthetic rates between the two species and among the different soil salinity levels, which are shown in Fig. R2d and Figs. R3a and d, respectively. Understanding the mangroves stomatal behavior relative to soil salinity and associated regulation in photosynthesis has not been well established from field data as discussed by Perri et al. (2019). Further field-based studies and implementation to the model are needed for better

representation of mangroves' stomatal conductance and associated regulation of photosynthesis under salt stress.

As an action for manuscript revision, we will include Fig. R1 in the Supporting Information, and the condensed version of the above descriptions on sensitivity analysis methodology and interpretations of the results in the method and discussion sections, respectively.

[Figure]

Figure R1: Sensitivity of above-ground biomass (AGB) of *R. stylosa* (*R. s*) and *B. gymnorrhiza* (*B. g*) across a soil salinity gradient to changes in parameter values of plant hydraulic traits: sensitivity to (a, b) salt filtration efficiency ($\varepsilon$), (c, d) water potential at which 50% of xylem conductivity is lost ($P_{50}$), (e, f) critical leaf water potential ($\psi_{lk}$) and minimum leaf water potential ($\psi_{l,min}$), and (g, h) sensitivity of marginal water use efficiency to leaf water potential ($\beta_0$). Sensitivities were examined by changing a value of one species (*R. s* or *B. g*) to the one determined for the other species shown in Table 2. Median

(solid line) and 90[th] percentile (shading) of AGB in steady states (> 300 years) are shown; the results are from one simulation without the ensemble approach.

RC1:
Second, compared with the plant hydraulics-salinity interaction, the efficacy of two other new modules - dynamic allocation and nutrient limitation is not well demonstrated. For example, Fig. 9 shows the huge plasticity of allometry in the model without much support from empirical data. Fig. S1 seems to suggest the allometric plasticity is observed but it is really hard to relate. Meanwhile, Fig.5 shows that including a more realistic DIN gradient did not improve the model results. Consider either including some more empirical supports or make them less central to the manuscript.

Response:
We admit that the simulated morphological traits and plasticity have not been sufficiently supported by observed data. The data shown in Fig. 9 (Comley and Mcguiness, 2005 and Komiyama et al., 2005) are the only data that we could find, and therefore we would like to make these results less central to the manuscript as suggested. Alternatively, we would like to keep the model description on the biomass allocation module and nutrient limitation in the materials & methods section in the manuscript because they are necessary for understanding the model prediction of the plants' responses to salt stress. For example, the decreased productivity under increased salt stress predicted by the model is related to the change in the biomass allocation pattern in addition to the regulation of stomatal conductance; the increase in salt stress led to increased biomass allocation to the stem and roots relative to leaves (as shown in Fig. 9), and this reduced the whole-plant photosynthesis and transpiration (which is scaled to nitrogen uptake rate) and increased carbon (through the stem and root respiration and root turnover) and nitrogen (through the root turnover) cost relative to unit leaf area, thereby reducing the productivity.

As an action for manuscript revision, we will remove sentences relevant to the implications of the dynamic biomass allocation and nutrient limitation from the abstract and concluding remarks. We will also remove Section 3.4 "Modeled morphological traits and effects of soil salinity" and 4.3 "Implications of the predicted morphological traits" from the manuscript. The contents of Figure 9 will be moved to the Supporting Information and the results will be shortly discussed in Section 4.2 "Soil salinity and interspecific competition shaping the forest structural variables" as the plants' morphological responses to salt stress and associated decrease in plant productivity. We believe that the results relevant to nutrient limitation are already not in the central focus of the manuscript (the results with more realistic DIN gradient are shown in Supporting Information Fig. S5 and are discussed shortly in the manuscript L. 548–564), thus further revision regarding this aspect is not necessary.

RC1:
Third, it is strange that no outputs from the new hydraulic module (e.g. leaf water potential diurnal cycle and seasonality) is presented, which is important to show the performance of the new plant hydraulics module.

Response:
We did not include the outputs from the hydraulic module (leaf water potential dynamics) due to lack of observed data that support model outputs – data on temporal dynamics of leaf water potential and the response to changes in soil salinity are remarkably scarce in the case of mangroves.

The panels in Fig. R2 show the seasonal variations in atmospheric variables, photosynthesis, and transpiration, that are shown in Fig. 4 in the manuscript, with addition of the simulated leaf water potential (at midday and predawn), which we consider replacing Fig. 4 with for the manuscript revision. The panels in Fig. R3 show the diurnal variations of the simulated photosynthesis, transpiration, and leaf water potential of the two species during summer and winter under two different soil salinity conditions (30 ‰ and 24 ‰).

The midday leaf water potential showed seasonal variations (Fig. R2f) as with the photosynthesis and transpiration (Fig. 2d and e). Due to the partial salt uptake of *R. stylosa* (as indicated by the lower $\varepsilon$ value of this species) that alleviates osmotic potential difference between the soil and plant, the predawn leaf water potential of *R. stylosa* was constantly higher than that of *B. gymnorrhiza* (Fig. R2f). With the combination of the lower $\varepsilon$, $\psi_{lk}$, $\psi_{l,min}$, and higher $\beta_0$ of *R. stylosa* (Table 2), this species showed larger magnitude of leaf water potential reduction and higher leaf-level transpiration rate during summer (June–August) compared to *B. gymnorrhiza* (Figs. R2e and f). Transpiration rates of both species decreased during winter (December–February), which resulted in the similar variations in midday leaf water potential of the two species. In contrast, leaf-level photosynthetic rates of the two species were at almost the same level throughout the year (Fig. R2d), suggesting that while the value of $\beta_0$ for *B. gymnorrhiza* was set to regulate stomatal conductance compared to *R. stylosa*, the stomatal regulation was not at the level that could significantly affect the photosynthetic rate.

Compared to salinity condition 24 ‰, both species showed significantly lowered leaf-level transpiration rates under salinity condition 30 ‰ especially during summer (Fig. R3b), suggesting the stomatal regulation of the transpiration and correspondingly the water (and nutrient) uptake from the soil under high soil salinity conditions. Alternatively, the decrease in leaf-level photosynthetic rates were not significant (Fig. R3a). The leaf water potential during night-time was lower when soil salinity was 30 ‰ compared to conditions when salinity was 24 ‰, due to the different osmotic potential in soil porewater, but the leaf water potential showed almost the same levels at midday during summer, which were close to the values of $\psi_{lk}$ determined for each species (Fig. R3c). The reduction in the leaf water potential to the level of $\psi_{lk}$ suggests the role of dynamic biomass allocation that adjusts the whole-tree transpiration demands and hydraulic conductivity in constraining the leaf water potential dynamics. In contrast, the diurnal dynamics in leaf water potential during winter showed similar magnitude of reduction of the water potential at midday between the two soil salinity conditions (Fig. R3f), suggesting that the atmospheric control on stomatal conductance and associated dynamics is more significant than the salinity control in winter.

As an action for manuscript revision, we will replace Fig. 4 with Fig. R2. We also include Fig. R3 in the manuscript after Fig. 4 (as Fig. 5). Short descriptions on the result interpretations related to the new figures (leaf water potential seasonal and diurnal dynamics with species and salinity differences) will be included in the result section. Discussions related to this revision will also be included in the discussion section.

[Figure]

Figure R2: Seasonal variations in atmospheric forcing variables: (a) solar radiation, (b) air temperature, and (c) vapor pressure deficit (VPD), and modeled seasonal dynamics: (d) monthly mean and standard deviation of gross photosynthetic rate ($P_g$, g C m$^{-2}$ ground day$^{-1}$) and (e) transpiration ($T$, mm day$^{-1}$) of *R. stylosa* and *B. gymnorrhiza* normalized with leaf layer index (LAI, m$^2$ leaf m$^{-2}$ ground) of the respective species, and (f) midday ($\psi_{l,midday}$) and predawn ($\psi_{l,predawn}$) leaf water potential of each species. Solar radiation is expressed as daily sum while air temperature and VPD are expressed as daily mean. Leaf water potential shown is the median value of individuals. Here, the modeled dynamics were from a simulation of soil salinity set as 30‰, and the results of the year when LAI reached 1.55 are shown. During this time, the LAI of *R. stylosa* and *B. gymnorrhiza* were 0.87 and 0.68, respectively. In panel "d", the seasonal variations in GPP/LAI measured by Okimoto et al. (2007) are also shown as reference, the data of which are from an area with LAI = 1.55 in Fukido mangrove forest in 2000–2001.

[Figure]

Figure R3: Simulated averaged diurnal variations in (a, d) photosynthesis and (b, e) transpiration of *R. stylosa* and *B. gymnorrhiza* normalized with LAI of the respective species, and (c, f) leaf water potential of the two species averaged for summer (June–August) and winter (December–February) under two soil salinity conditions (30 ‰ and 24 ‰). The variations under soil salinity 30 ‰ correspond to the results shown in Fig. R2. The variations under soil salinity 24 ‰ are from the results of a year that showed the same LAI (1.55). The diurnal variations in leaf water potential were derived based on the median value of individuals.

RC1: Finally, why not make the codes publicly available (line 579), especially given the new model is built on several other open-source models.

Response:

We will consider to upload the codes in GitHub and include the URL in the revised manuscript after clarifying copy rights and improving the code readability.

RC1:
Some minor comments along with the order of the text:

L. 26-27: maybe I missed it but which figure shows the self-thinning process supported by field data?

Response:
We referred to the decreasing tree density with increasing individual tree biomass patterns shown in Fig. 8 as the self-thinning process. The relevant discussions on this point can be found in L. 450–455. We will add a sentence explaining that the observed pattern in the tree density and increasing individual tree biomass is a result of the self-thinning process, which can be widely observed in mangrove forests (e.g., Deshar et al., 2012; Kamara et al., 2012; Khan et al., 2013).

RC1:
L. 120. Figure 1a, please use Mainland China or Fujian (the province) when all the other names are at province/prefecture level

Response:
We will correct the Figure 1a. Thank you for pointing it out.

RC1:
L. 171. Fig. 2 I remembered most physiological processes use daily time steps in the original SEIB-DGVM? So the photosyntheis module was rewritten to hourly time step in this study?

Response:
Yes, the photosynthesis model in SEIB-DGVM was replaced with the leaf flux model of Bonan et al. (2014) and photosynthesis was simulated with hourly time step in this study. The relevant descriptions can be found in L. 205–212 in the method section. The descriptions on the adaptation of the leaf flux model of Bonan et al. (2014) in this study have been provided in Note S4 in the Supporting Information.

RC1:
L. 184-186. I am not an expert on hydraulics in saline waters but is the osmotic potential also determined by temperature?

Response: Yes, the osmotic potential is also a function of (porewater) temperature. However, please note that sensitivity of the osmotic potential to change in temperature is significantly small compared to salinity because the osmotic potential is expressed using the temperature unit of Kelvin as described below. In this study, the osmotic potential in the soil ($\psi_\pi$, MPa) was computed using the following equation, which was adopted from Perri et al. (2019)
$$\psi_\pi = -CRi_vT_w \times 10^{-6}$$
where, $C$ is the salt concentration (mol m$^{-3}$), $R$ is the universal gas constant (J K$^{-1}$ mol$^{-1}$), $i_v$ is the van't Hoff coefficient, and $T_w$ is the water temperature (K). Assuming that NaCl is the dominant solute, $C$ is given by $C = sal \times \rho_w / 58.44$, where $sal$ is the salinity of porewater expressed in ‰, $\rho_w$ is the water

density (kg m$^{-3}$) and $i_v$ = 2. Because the variations in porewater temperature $T_w$ were not solved in this study, a constant value 298.15 K was given for the computation of the osmotic potential.

As an action for manuscript, we will include a description that the constant temperature value was used for the calculation of osmotic potential.

RC1:
L. 397. I guess tree size distribution is also available? Why not compare the simulated and observed size distribution in addition to mean DBH (maybe in supplementary)

Response:
Yes, the tree size distribution is available from the model outputs, however, the comparison with field data is complicated. Tree size distributions vary even at the same salinity condition, as implied from Fig. 7a of the variations in the mean DBH at the same salinity levels; thus, it is difficult to determine a representative tree size distribution for a given salinity condition from the field data. Therefore, a fair comparison of tree distribution between the model and data along with the salinity gradient is difficult. Alternatively, we believe that the comparisons for the mean DBH, AGB, and tree density–mean individual AGB relationship (Figs. 7 and 8) provide a sufficient assessment of model reproducibility of the forest structures along the soil salinity gradient.

RC1:
L. 405. Fig. 8. It seems the model generally underestimates tree density? Any explanations?

Response:
Yes, the model unfortunately underestimated tree density specifically in salinity conditions higher than 30 ‰ where _R. stylosa_ starts to dominate as seen in the Fig. 8. We consider that this was because of the prescribed DBH-maximum crown diameter ($D^*_{crown}$) allometric relationship of _R. stylosa_ shown in Fig. S1, which generally gives larger $D^*_{crown}$ compared to observed values for this species. The crown diameters of individuals basically determine the tree accommodation spaces, and therefore the overestimated crown diameter may have resulted in the underestimation of the tree density. Please see Supporting Information L. 29–45 for the reason of choice of the DBH-$D^*_{crown}$ relationship for _R. stylosa_. We expect that giving more realistic DBH-$D^*_{crown}$ relationship for _R. stylosa_ will improve the model prediction, and this could be addressed in future studies. A short discussion on this point has been provided in L. 455–460.

As an action for manuscript revision, we will add an explanation of the reason of the underestimation of tree density.

References

Bonan, G. B., Williams, M., Fisher, R. A., & Oleson, K. W. (2014). Modeling stomatal conductance in the earth system: linking leaf water-use efficiency and water transport along the soil–plant–atmosphere continuum. Geoscientific Model Development, 7(5), 2193–2222.

Comley, B. W., & McGuinness, K. A. (2005). Above-and below-ground biomass, and allometry, of four common northern Australian mangroves. Australian Journal of Botany, 53(5), 431–436.

Deshar, R., Sharma, S., Mouctar, K., Wu, M., Hoque, A. T. M. R., & Hagihara, A. (2012). Self-thinning exponents for partial organs in overcrowded mangrove *Bruguiera gymnorrhiza* stands on Okinawa Island, Japan. Forest ecology and management, 278, 146-154.

Jiang, G. F., Goodale, U. M., Liu, Y. Y., Hao, G. Y., & Cao, K. F. (2017). Salt management strategy defines the stem and leaf hydraulic characteristics of six mangrove tree species. Tree physiology, 37(3), 389–401.

Kamara, M., Deshar, R., Sharma, S., Kamruzzaman, M. D., & Hagihara, A. (2012). The self-thinning exponent in overcrowded stands of the mangrove, *Kandelia obovata*, on Okinawa Island, Japan. Journal of oceanography, 68(6), 851-856.

Khan, M. N. I., Sharma, S., Berger, U., Koedam, N., Dahdouh-Guebas, F., & Hagihara, A. (2013). How do tree competition and stand dynamics lead to spatial patterns in monospecific mangroves?. Biogeosciences, 10(4), 2803–2814.

Komiyama, A., Poungparn, S., & Kato, S. (2005). Common allometric equations for estimating the tree weight of mangroves. Journal of tropical ecology, 21(4), 471–477.

Perri, S., Katul, G. G., & Molini, A. (2019). Xylem–phloem hydraulic coupling explains multiple osmoregulatory responses to salt stress. New Phytologist, 224(2), 644–662.

Reef, R., & Lovelock, C. E. (2015). Regulation of water balance in mangroves. Annals of botany, 115(3), 385–395.

---

## Author Comment (AC3)

Response to Referee #2 comment (RC2)

We would like to thank the reviewers for taking the time to review our manuscript and providing constructive comments. Please see our responses to the comments below.

RC2:
Yoshikai et al. present a well-articulated analysis of a model development effort centered on capturing mangrove ecosystem structure and long-term carbon storage using the individual based dynamic vegetation model SEIB-DGVM with a newly incorporated plant hydraulics model following Xu et al. 2016 and a salinity regulation component following the theoretical works of Perri et al. 2018 and 2019. Impressively, the new mangrove function model also accounts for the influence of nutrient availability (specifically nitrogen) alongside plant hydraulics. The new model proved capable of convincingly reproducing the behaviors of two species of mangrove along a soil salinity gradient in Japan. On the whole, the manuscript presents a strong, timely, and necessary contribution to DGVM and Earth system modeling, given the unique dynamics of mangrove ecosystems and their outsized influence on the carbon cycle. I have only minor questions and suggestions for the authors as they ready their work for publication.

Response:
Thank you very much for the positive assessment and constructive comments on our manuscript. We have addressed the comments suggested by the reviewer, as follows.

RC2:
L160: The introduction of an aboveground root biomass carbon pool is a particularly useful addition to this model and other mangrove/cypress systems. I am curious how the aboveground root biomass was accounted for allometricaly? Was this related more strongly with stem or crown diameters?

Response:
In the previous study of Yoshikai et al. (2021), the above-ground root structures of *Rhizophora stylosa* were extensively measured in our study site (Fukido mangrove forest), and an empirical model to predict the root morphology was established. The empirical model uses only DBH (diameter at breast height) as the explanatory variable. We used this model to compute the above-ground root biomass pool in the simulation, therefore it is related only to DBH. The data and empirical model prediction of DBH and above-ground root volume relationship are provided in Fig. R1.
As an action for manuscript revision, we will add an explanation about this point.

[Figure]

Figure R1. Relationship between DBH and above-ground root volume of *R. stylosa* measured in Fukido mangrove forest. An empirical model developed in Yoshikai et al. (2021) was used for the prediction.

RC2:
L195: Was the sapwood allometric relationship specific to the two mangrove species simulated in this study or is this a general equation?

Response:
We consider that this is a general equation. In the work of Trugman et al. (2019) on terrestrial ecosystems, they applied this kind of simplified relationship for estimating sapwood area and stated that the simplification is broadly consistent with reports in literature.

RC2:
L203: How was LAI measured in this study? Were different values used for the different species?

Response:
LAI has not been measured in the study site.
In L203, the leaf area, $LA$ (m$^2$ tree$^{-1}$), is calculated using $M_L \times SLA \times 10^{-4}$, where $M_L$ is the leaf biomass (g tree$^{-1}$), which is a state variable in the model, and $SLA$ is the specific leaf area (cm$^2$ g$^{-1}$), which is a model parameter as shown in Table 2.
As an action for manuscript revision, we will include the statement that the simulated LAI has not been validated and will be considered for future study.

RC2:
Table 1: It looks like there are a few sources missing (e.g. $D_{crown,con}$ and $H_{con}$) what values were used for these and were they assumed or developed from literature or field observation?

Response:

Physical constraint on crown diameter ($D_{\text{crown,con}}$) and on tree height ($H_{\text{con}}$) are the variables to be computed in the SEIB-DGVM based on the relative distances of a tree with surrounding trees as illustrated in Figs. 3c–d. Therefore, there is no source and value to add in Table 1 for these variables. For the parameter $\beta_{\text{stock}}$ (Target C and N in the stock pool relative to the stem), we gave an assumed value because we were not able to find any value from literature. The given value (0.05) is comparable to the value of model prediction of nonstructural carbohydrate reserves (NSC) and xylem biomass ratio in Trugman et al. (2018).

As an action for manuscript revision, we will indicate in Table 1 that the value for $\beta_{\text{stock}}$ is an assumed value.

RC2:
L240: How were the values for critical leaf water potential determined? Were these values optimized?

Response:
Yes, the value for the critical leaf water potential was optimized for each species so that the simulated above-ground biomass (AGB) and mean DBH agree with the field data.
Relevant descriptions can be found in L312–314.

RC2:
L350: It would be useful to restate the present-day average salinity for comparison's sake.

Response:
Thank you for the suggestion. We will add a sentence after L. 350 that states that the present-day average salinity of the survey plots is 28 ‰.

RC2:
L367: More discussion of the simulated *B. gymnorrhiza* mortality would be useful and interesting. Was there a programmed lifespan that triggered this event?

Response:
L367 refers to the deaths of large *B. gymnorrhiza* trees that generated forest gaps and promoted the establishment of small trees under 20 ‰ salinity condition.
We did not prescribe any lifespan in the model due to the lack of knowledge on mangroves' longevity (the original SEIB-DGVM defines tree longevity as a model parameter, but this factor was excluded in this study for this reason; the relevant description can be found in L100–101 in the Supporting Information). We also did not introduce any specific processes for the death of large trees such as size- and age-dependent mortality (please see Note S3 in the Supporting Information for the processes related to tree mortality). Therefore, tree deaths occur without dependence on tree size or age in the simulation. On the other hand, only deaths of large trees generated forest gaps that promoted tree establishment because the deaths of small trees resulted in growth stimulation of the surrounding trees and the space created by the death were eventually filled by the canopy of the surrounding trees; this process can be seen in Fig. 8 showing the decreasing tree density with increasing individual tree biomass.
In this regard, we figured out that the term "onset of deaths of large *B. gymnorrhiza* trees" in L367 was confusing because it seems as if some specific processes triggered large trees' mortality. Instead,

we should have written it as "onset of formation of forest gaps resulted from deaths of large *B. gymnorrhiza* trees" to convey our intention correctly. We are sorry about it.

As an action for manuscript revision, we will correct this point.

RC2:

L394: A figure citation here where this comparison is shown would be helpful.

Response:

We will add figure citation (which is Fig. 7b). Thank you for the suggestion.

RC2:

L510: At what time increment is the optimization of the DBH-H adjustment applied?

Response:

The morphological adjustment occurs at daily time step by the flexible biomass allocation as indicated in Fig. 2. The detailed procedures on biomass allocation are described in Fig. 3.

RC2:

L521: Does the increase in root biomass refer to both above and below ground roots or are the aboveground roots lumped into the shoot category in this scenario?

Response:

The statement in L521 refers to the result shown in Fig. 9. In the figure, the root biomass refers to the sum of coarse root and fine root biomass while the shoot biomass refers to the sum of stem and leaf biomass. Thus, above-ground roots are not accounted for in both shoot and root biomass in Fig. 9. The description about this point has been provided in the caption of Fig. 9 (L422–430).

RC2:

L579: It would be nice to see the code released with a DOI in a Zenodo repository or the like given the relevance of this modeling effort to the broader community of models.

Response:

We will consider to upload the codes in GitHub and include the URL in the revised manuscript after clarifying copy rights and improving the code readability. Thank you for your interest.

RC2:

Finally, there are a few instances of minor grammatical errors (subject-verb agreement and plurals versus possessives) that could be addressed through the use of a grammar editing service.

Response:

We are sorry about the grammatical errors. We will carefully check the manuscript again for the revision (this manuscript is actually a version after an English proofing service).

References

Trugman, A. T., Anderegg, L. D., Wolfe, B. T., Birami, B., Ruehr, N. K., Detto, M., ... & Anderegg, W. R. (2019). Climate and plant trait strategies determine tree carbon allocation to leaves and mediate future forest productivity. Global change biology, 25(10), 3395-3405.

Trugman, A. T., Detto, M., Bartlett, M. K., Medvigy, D., Anderegg, W. R. L., Schwalm, C., ... & Pacala, S. W. (2018). Tree carbon allocation explains forest drought-kill and recovery patterns. Ecology Letters, 21(10), 1552-1560.

Yoshikai, M., Nakamura, T., Suwa, R., Argamosa, R., Okamoto, T., Rollon, R., ... & Nadaoka, K. (2021). Scaling relations and substrate conditions controlling the complexity of *Rhizophora* prop root system. Estuarine, Coastal and Shelf Science, 248, 107014.

---

## Author Response (AR1)

Dear editor and reviewers,

We thank the reviewers for taking time to review our manuscript and provide constructive comments. Below we address the reviewers' comments point by point. We also would like to note that some minor points have been corrected as follows, which were pointed out by the editorial support team.

1. The reference list has been corrected to suit journal standards.
2. The color schemes of Figures 5, 6, 9 (now Figs. 6, 7, and S7, respectively, in the revised manuscript), and S1 have been changed to a color vision deficiency-friendly scheme.

**Response to Referee #1 comment (RC1)**

**RC1:**

Summary

Yoshikai et al. provide a novel modeling study to understand and predict mangrove forest dynamics across a soil salinity gradient. The study added a plant hydraulic module, dynamic allocation module, and nutrient (nitrogen) limitation on growth into SEIB-DGVM. The new model allows for consideration of soil salinity effects on plant ecophysiology as well as soil nutrient levels in mangrove forests. After calibration of two parameters determining allocation and stomata that are unavailable from literature, the model can well represent the spatial gradient of forest structure (mean DBH) and biomass (AGB) across a salinity gradient in a mangrove forest in Japan. Other model-data comparison is also presented. Altogether, the authors conclude that including hydraulic trade-offs and differences in the ability to deal with salinity is critical and adequate for predicting dominant forest dynamics in mangrove forests.

Comments

I really like the study, which extends the existing plant hydrodynamic modeling framework (often used and calibrated in arid/semi-arid ecosystems) to coastal saline ecosystems (also water-stressed). The idea of plant hydraulic control on mangrove forest dynamics existed for some time but the study presents a novel modeling study to evaluate the idea together with field data. Overall, the manuscript is well written and includes adequate details for understanding the model. I have three major comments about model diagnostics, which hopefully can improve the manuscript.

**Response:**

We thank the reviewer for the positive assessment and constructive comments on our manuscript. We have addressed the comments suggested by the reviewer as follows.

**RC1:**

First, in my opinion, the key evidence to the manuscript's conclusion is Fig. 7&8, which shows how simulated forest structure and biomass match with observed values across the salinity gradient after only modest model tuning (2 parameters in Table 2). However, it is always more important and interesting to know why the model can reproduce the observations. What trait/parameters/processes are dominant in driving the model output. Is it salt filtration efficiency? $P_{50}$?, $\psi_{lk}$?, or $\beta_0$? I would suggest running some sensitivity tests to show what traits/parameters lead to the pattern in Fig.7 and how important is the tuning of $\psi_{lk}$ and $\beta_0$ (their differences seem to be small). In fact, I am curious about whether salt filtration efficiency or $P_{50}$ is more important, or maybe they have to be coordinated in the model to explain the observed pattern. Such information will make the study more useful.

**Response:**

We have conducted sensitivity analysis of the plant hydraulic trait parameters ($\varepsilon$, $P_{50}$, $\psi_{lk}$, and $\beta_0$) to see the relative importance of each parameter in reproducing the observed pattern of the forest structure across a soil salinity gradient as suggested. We specifically looked into the sensitivity of above-ground biomass (AGB), which showed contrasting changes of the two species (*Rhizophora stylosa* and *Bruguiera gymnorrhiza*) to changes in soil salinity in the forest (as shown in Fig. 7b in the original manuscript). Please note that to examine the sensitivity of $\psi_{lk}$, we changed the values of both $\psi_{lk}$ and $\psi_{l,min}$ to keep the buffer between the two parameter values; a decrease in $\psi_{lk}$ without decrease in $\psi_{l,min}$ may otherwise lead to the xylem water conductance becoming more susceptible to the water potential at which the stomata closes.

For the analysis, we changed the value of a target parameter of one species (either *R. stylosa* or *B. gymnorrhiza*) to the one determined for the other species as shown in Table 2, and ran the "salinity gradient simulation". The results are shown in Fig. R1 in this document. To save on computational cost, we ran only one simulation for each sensitivity test instead of the ensemble approach done for reproducing the forest structures as shown in Fig. 7 in the original manuscript. The omission of ensemble runs resulted in some fluctuations in AGB along the soil salinity gradient (Fig. R1). However, the fluctuations were not at a level that could affect the interpretation of the overall simulated forest structural patterns across the soil salinity gradient.

The results showed that the change in the values of the parameters $\psi_{lk}$ and $\psi_{l,min}$ had the most impact on the simulation results. The decreases in $\psi_{lk}$ and $\psi_{l,min}$ of *B. gymnorrhiza* to the level determined for *R. stylosa* largely increased the salt tolerance of this species and resulted in the *B. gymnorrhiza*-dominated forest even at high soil salinity conditions (i.e., > 34 ‰) (Fig. R1f). On the other hand, the increase in these parameters for *R. stylosa* to the level for *B. gymnorrhiza* reduced the salt stress tolerance of *R. stylosa* and resulted in the unsuccessful growth of this species even at soil salinity higher than 30 ‰ where *R. stylosa* starts to dominate in the forest (Fig. R1e). These results indicate that the mangroves' capacity in reducing the leaf water potential is one of the most important functional traits characterizing their salt tolerance as suggested by Reef and Lovelock (2015). The response of

AGB to changes in $\psi_{lk}$ also indicates the substantial impact of biomass allocation dynamics determined by $\psi_{lk}$ on plant productivity.

The parameter that has impacted the simulation results next to $\psi_{lk}$ and $\psi_{l,min}$ was the salt filtration efficiency, $\varepsilon$ (Figs. R1a–b). The results shown in Figs. R1a–b highlighted the benefit of partial uptake of salt and associated reduction in xylem tension of *R. stylosa* to maintain productivity under relatively high soil salinity (i.e., > 32 ‰). The changes in the values of $P_{50}$ also affected the simulation results to some extent (Figs. R1c–d). The increase in $P_{50}$ of *B. gymnorrhiza*, which increases the vulnerability to xylem cavitation, decreased the productivity of this species, and resulted in *R. stylosa*-dominated forest at soil salinity 30 ‰ where the two species showed the same level of AGB in the simulation result shown in Fig. 7 (in the original manuscript). The decrease in $P_{50}$ of *R. stylosa* increased the AGB of this species by around 15 Mg ha$^{-1}$ compared to the case shown in Fig. 7. While the model demonstrated relatively high sensitivities to these parameters ($\varepsilon$ and $P_{50}$), it is considered that these are coordinated functional traits, i.e., the lower cavitation resistance (as indicated by higher $P_{50}$ of *R. stylosa*) may result from incomplete salt removal (as indicated by higher $\varepsilon$ of *R. stylosa*) that reduces xylem tension required to maintain water uptake (Jiang et al., 2017). Therefore, they may have to be defined as coordinated plant functional traits resulting from adaptation to salt stress in the model.

The sensitivity of AGB to $\beta_0$ turned out to be quite low suggesting that the choice of -0.6 for $\beta_0$ already leads to efficient stomatal openings for photosynthesis compared to the case of -0.4 for $\beta_0$ (Fig. R1g–h, Table 2). This may explain the small variations in the leaf-level photosynthetic rates between the two species and among the different soil salinity levels, which are shown in Fig. R2d and Figs. R3a and d in this document, respectively. Understanding the mangroves' stomatal behavior relative to soil salinity and covariation with leaf water potential and photosynthesis have not been well established from field data as discussed by Perri et al. (2019). Further field-based studies and data implementation to the model are needed for better representation of mangroves' stomatal conductance and associated regulation of photosynthesis under salt stress.

In the revised manuscript, we have included Fig. R1 in the Supporting Information as Fig. S6, and the condensed version of the above descriptions of the methods used for sensitivity analysis and the interpretations of results in the method (L. 318–324 or L. 329–335 in marked-up version) and discussion sections (L. 506–522 or L. 544–560 in marked-up version), respectively.

[Figure]

Figure R1: Sensitivity of the above-ground biomass (AGB) of *R. stylosa* (*R. s*) and *B. gymnorrhiza* (*B. g*) across a soil salinity gradient to changes in parameter values of plant hydraulic traits: sensitivity to (a, b) salt filtration efficiency ($\varepsilon$), (c, d) water potential at which 50% of xylem conductivity is lost ($P_{50}$), (e, f) critical leaf water potential ($\psi_{lk}$) and minimum leaf water potential ($\psi_{l,min}$), and (g, h) sensitivity of marginal water use efficiency to leaf water potential ($\beta_0$). Sensitivities were examined by changing the value of one species (*R. s* or *B. g*) to the one determined for the other species shown in Table 2. Median (solid line) and 90th percentile (shading) of AGB in steady states (> 300 years) are shown; results are from one simulation without the ensemble approach.

**RC1:**

Second, compared with the plant hydraulics-salinity interaction, the efficacy of two other new modules - dynamic allocation and nutrient limitation is not well demonstrated. For example, Fig. 9 shows the huge plasticity of allometry in the model without much support from empirical data. Fig. S1 seems to suggest the allometric plasticity is observed but it is really hard to relate. Meanwhile, Fig.5 shows that including a more realistic DIN gradient did not improve the model results. Consider either including some more empirical supports or make them less central to the manuscript.

**Response:**

We believe that the reviewer was referring to Fig. S5, not Fig. 5.

We admit that the simulated morphological traits and plasticity have not been sufficiently supported by observed data. The data shown in Fig. 9 (Comley and Mcguiness, 2005 and Komiyama et al., 2005) are the only data that we could find, and therefore we have made these results less central to the manuscript as suggested. Nevertheless, we kept the model description on the biomass allocation module and nutrient limitation in the materials & methods section in the revised manuscript because they are necessary for understanding the model prediction of the plants' responses to salt stress. For example, the decreased productivity under increased salt stress predicted by the model is related to the change in biomass allocation pattern in addition to the regulation of stomatal conductance; the increase in salt stress led to increased biomass allocation to the stem and roots relative to leaves (as shown in Fig. 9 in the original manuscript, now in Fig. S7), and this reduced the whole-plant photosynthesis and transpiration (which is scaled to nitrogen uptake rate) and increased carbon (through the stem and root respiration and root turnover) and nitrogen (through the root turnover) cost relative to unit leaf area, thereby reducing the productivity.

In the revised manuscript, we have removed sentences relevant to the implications of the dynamic biomass allocation and nutrient limitation from the abstract and concluding remarks (as seen in L. 28–29, and L. 659–660 in the marked-up version, respectively). We also removed Section 3.4 "Modeled morphological traits and effects of soil salinity" and 4.3 "Implications of the predicted morphological traits" from the manuscript (as seen in L. 460–470, and L. 583–614 in the marked-up version), and few sentences about the implications of nutrient limitation from Section 4.4 (as seen in L. 631–634 in the marked-up version). The contents of Figure 9 in the original manuscript have been moved to the Supporting Information as Figure S7, and the results are discussed briefly in Section 4.2 "Soil salinity and interspecific competition shaping the forest structural variables" as the plants' morphological responses to salt stress and associated decrease in plant productivity (L. 496–505, or L. 534–543 in the marked-up version). We believe that the results relevant to nutrient limitation are already not the central focus of the manuscript (the results with more realistic DIN gradient are shown in Supporting Information Fig. S5 and are discussed shortly in L. 560–575 of the revised version of the manuscript, or L. 635–650 of the marked-up version), thus further

revision regarding this aspect has not been made except the removal of few sentences from Section 4.4 (L. 631–634 in the marked-up version).

**RC1:**

Third, it is strange that no outputs from the new hydraulic module (e.g. leaf water potential diurnal cycle and seasonality) is presented, which is important to show the performance of the new plant hydraulics module.

**Response:**

We did not include the outputs from the hydraulic module (leaf water potential dynamics) due to lack of observed data that support model outputs; data on temporal dynamics of leaf water potential and the response to changes in soil salinity are remarkably scarce in the case of mangroves.

The panels in Fig. R2 in this document show the seasonal variations in atmospheric variables, photosynthesis, and transpiration shown in Fig. 4 in the original manuscript, with addition of the simulated leaf water potential (at midday and predawn), which we have replaced Fig. 4 with in the revised manuscript. The panels in Fig. R3 show the diurnal variations of the simulated photosynthesis, transpiration, and leaf water potential of the two species during summer and winter under two different soil salinity conditions (30 ‰ and 24 ‰).

The midday leaf water potential showed seasonal variations (Fig. R2f) like photosynthesis and transpiration (Fig. R2d and e). Due to the partial salt uptake of *R. stylosa* (as indicated by the lower $\varepsilon$ value of this species; Table 2) that alleviates osmotic potential difference between the soil and plant, the predawn leaf water potential of *R. stylosa* was constantly higher than that of *B. gymnorrhiza* (Fig. R2f). With the combination of lower $\varepsilon$, $\psi_{lk}$, $\psi_{l,min}$, and higher $\beta_0$ of *R. stylosa* (Table 2), this species showed larger magnitude of leaf water potential reduction and higher leaf-level transpiration rate during summer (June–August) compared to *B. gymnorrhiza* (Figs. R2e and f). Transpiration rates of both species decreased during winter (December–February), which resulted in the similar variations in midday leaf water potential of the two species. In contrast, leaf-level photosynthetic rates of the two species were at almost the same level throughout the year (Fig. R2d), suggesting that while the value of $\beta_0$ for *B. gymnorrhiza* was set to regulate stomatal conductance compared to *R. stylosa*, the stomatal regulation was not at the level that could significantly affect the leaf-level photosynthetic rate.

Compared to salinity condition 24 ‰, both species showed significantly lowered leaf-level transpiration rates under salinity condition 30 ‰ especially during summer (Fig. R3b), suggesting the stomatal regulation of transpiration and correspondingly, the water (and nutrient) uptake from the soil under high soil salinity conditions. On the other hand, the decrease in leaf-level photosynthetic rates were not significant (Fig. R3a). The leaf water potential during night-time was lower when soil salinity was 30 ‰ compared to conditions

when salinity was 24 ‰, due to the different osmotic potential in soil porewater. The leaf water potential at midday, however, showed almost the same levels during summer, which were close to the values of $\psi_{lk}$ determined for each species (Fig. R3c, Table 2). The reduction in leaf water potential to the level of $\psi_{lk}$ suggests the role of dynamic biomass allocation that adjusts the whole-tree transpiration demands and hydraulic conductivity in constraining the leaf water potential dynamics. In contrast, the diurnal dynamics in leaf water potential during winter showed similar magnitude of reduction of the water potential at midday between the two soil salinity conditions (Fig. R3f), suggesting that the atmospheric control on stomatal conductance and associated dynamics is more significant than the salinity control in winter.

      In the revised manuscript, we have replaced Fig. 4 with Fig. R2. We also included Fig. R3 after Fig. 4 (as Fig. 5). Short descriptions on the result interpretations related to the new figures (leaf water potential seasonal and diurnal dynamics with species and salinity differences) have been included in the result section (L. 346–352 and 363–374, or L. 358–364 and 376–387 in the marked-up version). Discussions related to this revision have also been included in the discussion section (in L. 496–522 and 553–556, or in L. 535–560 and 624–626 in the marked-up version).

[Figure]

Figure R2: Seasonal variations in atmospheric forcing variables: (a) solar radiation, (b) air temperature, and (c) vapor pressure deficit (VPD), and modeled seasonal dynamics: (d) monthly mean and standard deviation of gross photosynthetic rate ($P_g$, g C m$^{-2}$ ground day$^{-1}$) and (e) transpiration ($T$, mm day$^{-1}$) of *R. stylosa* and *B. gymnorrhiza* normalized with leaf layer index (LAI, m$^2$ leaf m$^{-2}$ ground) of the respective species, and (f) midday ($\psi_{l,midday}$) and predawn ($\psi_{l,predawn}$) leaf water potential of each species. Solar radiation is expressed as daily sum while air temperature and VPD are expressed as daily mean. Leaf water potential shown is the median value of individuals. Here, the modeled dynamics were from a simulation of soil salinity set as 30 ‰, and the results of the year when LAI reached 1.55 are shown. During this time, the LAI of *R. stylosa* and *B. gymnorrhiza* were 0.87 and 0.68, respectively. In panel "d", the seasonal variations in $P_g$/LAI measured by Okimoto et al. (2007) are also shown as reference, the data of which are from an area with LAI = 1.55 in Fukido mangrove forest in 2000–2001.

[Figure]

Figure R3: Simulated averaged diurnal variations in (a, d) photosynthesis and (b, e) transpiration of *R. stylosa* and *B. gymnorrhiza* normalized with LAI of the respective species, and (c, f) leaf water potential of the two species averaged for summer (June–August) and winter (December–February) under two soil salinity conditions (30 ‰ and 24 ‰). The variations under soil salinity 30 ‰ correspond to the results shown in Fig. R2. The variations under soil salinity 24 ‰ are from the results of a year that showed the same LAI (1.55). The diurnal variations in leaf water potential were derived based on the median value of individuals.

RC1:

Finally, why not make the codes publicly available (line 579), especially given the new model is built on several other open-source models.

**Response:**
We have uploaded the model code in the following link with permission of Dr. Hisashi Sato, the developer of the SEIB-DGVM. We revised the section "Code Availability" correspondingly. https://github.com/MasayaYoshikai/SEIB_mangrove

**RC1:**
Some minor comments along with the order of the text:

L. 26-27: maybe I missed it but which figure shows the self-thinning process supported by field data?

**Response:**
We referred to the decrease in tree density with increase in individual tree biomass patterns shown in Fig. 8 (now Fig. 9 in the revised manuscript) as the self-thinning process. The relevant discussions about this point have been provided in the original manuscript (L. 450–455, or 463–468 in the revised manuscript).
We added a sentence in the revised manuscript explaining that the observed pattern in tree density and increasing individual tree biomass is a result of the self-thinning process, which can be widely observed in mangrove forests (e.g., Deshar et al., 2012; Kamara et al., 2012; Khan et al., 2013); this could be seen in L. 463–465 in the revised manuscript, or L. 500 –502 in the marked-up version.

**RC1:**
L. 120. Figure 1a, please use Mainland China or Fujian (the province) when all the other names are at province/prefecture level

**Response:**
We have corrected Figure 1a as suggested. We thank the reviewer for pointing it out.

**RC1:**
L. 171. Fig. 2 I remembered most physiological processes use daily time steps in the original SEIB-DGVM? So the photosyntheis module was rewritten to hourly time step in this study?

**Response:**
Yes, the photosynthesis model in SEIB-DGVM was replaced with the leaf flux model of Bonan et al. (2014) and photosynthesis was simulated in hourly time step in this study. The relevant

descriptions have been provided in L. 205–212 of the method section in the original manuscript, or in L. 203–207 in the revised manuscript. The descriptions on the adaptation of the leaf flux model of Bonan et al. (2014) in this study have been provided in Note S4 in the Supporting Information.

**RC1:**

L. 184-186. I am not an expert on hydraulics in saline waters but is the osmotic potential also determined by temperature?

**Response:**

Yes, the osmotic potential is also a function of (porewater) temperature. However, please note that the sensitivity of osmotic potential to change in temperature is significantly small compared to salinity because the osmotic potential is expressed using the temperature unit of Kelvin as described below. In this study, the osmotic potential in the soil ($\psi_\pi$, MPa) was computed using the following equation, which was adopted from Perri et al. (2019)

$$\psi_\pi = -CRi_vT_w \times 10^{-6}$$

where, $C$ is the salt concentration (mol m$^{-3}$), $R$ is the universal gas constant (J K$^{-1}$ mol$^{-1}$), $i_v$ is the van't Hoff coefficient, and $T_w$ is the water temperature (K). Assuming that NaCl is the dominant solute, $C$ is given by $C = sal \times \rho_w/58.44$, where $sal$ is the salinity of porewater expressed in ‰, $\rho_w$ is the water density (kg m$^{-3}$) and $i_v = 2$. Because the variations in porewater temperature $T_w$ were not solved in this study, a constant value 298.15 K was given for the computation of the osmotic potential.

We have included a description that the constant temperature value was used for the calculation of osmotic potential in L. 182–184 in the revised manuscript, or L. 188–190 in the marked-up version.

**RC1:**

L. 397. I guess tree size distribution is also available? Why not compare the simulated and observed size distribution in addition to mean DBH (maybe in supplementary)

**Response:**

Yes, the tree size distribution is available from the model outputs, however, the comparison with field data is complicated. Tree size distributions vary even at the same salinity condition as implied by the variations in the mean DBH at the same salinity levels in Fig. 7a (now Fig. 8a in the revised manuscript); thus, it is difficult to determine a representative tree size distribution for a given salinity condition from the field data. Therefore, a fair comparison of tree distribution between the model and data along with the salinity gradient is difficult. Alternatively, we believe that the comparisons for the mean DBH, AGB, and tree density–mean individual AGB relationship (Figs. 7 and 8, or Figs. 8 and 9 in the revised manuscript)

provide a sufficient assessment of the model reproducibility of forest structures along the soil salinity gradient.

**RC1:**
L. 405. Fig. 8. It seems the model generally underestimates tree density? Any explanations?

**Response:**
Yes, the model unfortunately underestimated tree density specifically in salinity conditions higher than 30 ‰ where *R. stylosa* starts to dominate as seen in Fig. 8 (now Fig. 9). We consider that this was because of the prescribed DBH-maximum crown diameter ($D^*_{crown}$) allometric relationship of *R. stylosa* shown in Fig. S1, which generally gives larger $D^*_{crown}$ compared to observed values for this species. The crown diameters of individuals basically determine the tree accommodation spaces, and therefore the overestimated crown diameter may have resulted in the underestimation of the tree density. Please see Supporting Information L. 29–45 for the reason of choice of the DBH-$D^*_{crown}$ relationship for *R. stylosa*. We expect that giving more realistic DBH-$D^*_{crown}$ relationship for *R. stylosa* will improve the model prediction, and this could be addressed in future studies. A short discussion on this point has been provided in L. 455–460 in the original manuscript or L. 469–479 in the revised manuscript.

In the revised manuscript, we added an explanation of the reason of the underestimation of tree density (L. 469–473, or L. 507–512 in the marked-up version).

**Response to Referee #2 comment (RC2)**

**RC2:**
Yoshikai et al. present a well-articulated analysis of a model development effort centered on capturing mangrove ecosystem structure and long-term carbon storage using the individual based dynamic vegetation model SEIB-DGVM with a newly incorporated plant hydraulics model following Xu et al. 2016 and a salinity regulation component following the theoretical works of Perri et al. 2018 and 2019. Impressively, the new mangrove function model also accounts for the influence of nutrient availability (specifically nitrogen) alongside plant hydraulics. The new model proved capable of convincingly reproducing the behaviors of two species of mangrove along a soil salinity gradient in Japan. On the whole, the manuscript presents a strong, timely, and necessary contribution to DGVM and Earth system modeling, given the unique dynamics of mangrove ecosystems and their outsized influence on the carbon cycle. I have only minor questions and suggestions for the authors as they ready their work for publication.

**Response:**

We thank the reviewer for the positive assessment and constructive comments on our manuscript. We have addressed the comments suggested by the reviewer as follows.

**RC2:**

L160: The introduction of an aboveground root biomass carbon pool is a particularly useful addition to this model and other mangrove/cypress systems. I am curious how the aboveground root biomass was accounted for allometricaly? Was this related more strongly with stem or crown diameters?

**Response:**

In the previous study of Yoshikai et al. (2021), the above-ground root structures of *Rhizophora stylosa* were extensively measured in our study site (Fukido mangrove forest), and an empirical model to predict the root morphology was established. The empirical model uses only DBH (diameter at breast height) as the explanatory variable. We used this model to compute the above-ground root biomass pool in the simulation, therefore it is related only to DBH. The data and empirical model prediction of DBH and above-ground root volume relationship are provided in Fig. R4 in this document.

We have added an explanation about this point in the revised manuscript (L. 252–253, or L. 260–261 in the marked-up version).

[Figure]

Figure R4. Relationship between DBH and above-ground root volume of *R. stylosa* measured in Fukido mangrove forest. An empirical model developed in Yoshikai et al. (2021) was used for the prediction.

**RC2:**

L195: Was the sapwood allometric relationship specific to the two mangrove species simulated in this study or is this a general equation?

**Response:**

We consider that this is a general equation. In the work of Trugman et al. (2019) on terrestrial ecosystems, they applied this kind of simplified relationship for estimating sapwood area and stated that the simplification is broadly consistent with reports in literature.

**RC2:**

L203: How was LAI measured in this study? Were different values used for the different species?

**Response:**

LAI has not been measured in the study site.

In L. 203, the leaf area, $LA$ (m$^2$ tree$^{-1}$), is calculated using $M_L \times SLA \times 10^{-4}$, where $M_L$ is the leaf biomass (g tree$^{-1}$), which is a state variable in the model, and $SLA$ is the specific leaf area (cm$^2$ g$^{-1}$), which is a model parameter as shown in Table 2.

In the revised manuscript, we have included the statement in the discussion section that the simulated LAI has not been validated and will be considered for future study (L. 476–479, or L. 515–517 in the marked-up version).

**RC2:**

Table 1: It looks like there are a few sources missing (e.g. $D_{crown,con}$ and $H_{con}$) what values were used for these and were they assumed or developed from literature or field observation?

**Response:**

Physical constraint on crown diameter ($D_{crown,con}$) and on tree height ($H_{con}$) are the variables to be computed in the SEIB-DGVM based on the relative distances of a tree with surrounding trees as illustrated in Figs. 3c–d. Therefore, there is no source and value to add in Table 1 for these variables. For the parameter $\beta_{stock}$ (Target C and N in the stock pool relative to the stem), we used an assumed value because we were not able to find any value from literature. The given value (0.05) is comparable to the value of model prediction of nonstructural carbohydrate reserves (NSC) and xylem biomass ratio in Trugman et al. (2018).

In the revised manuscript, we indicated in Table 1 that the value for $\beta_{stock}$ is an assumed value.

**RC2:**

L240: How were the values for critical leaf water potential determined? Were these values optimized?

**Response:**

Yes, the value for the critical leaf water potential was optimized for each species so that the simulated above-ground biomass (AGB) and mean DBH agree with the field data.

Relevant descriptions have been provided in L.312–314 in the original manuscript or L. 310–312 in the revised manuscript.

**RC2:**

L350: It would be useful to restate the present-day average salinity for comparison's sake.

**Response:**

We thank the reviewer for the suggestion.

In the revised manuscript, we have added a sentence that states that the present-day average salinity of the survey plots is 28 ‰ (L. 384–385, or L. 397–398 in the marked-up version).

**RC2:**

L367: More discussion of the simulated *B. gymnorrhiza* mortality would be useful and interesting. Was there a programmed lifespan that triggered this event?

**Response:**

L367 refers to the deaths of large *B. gymnorrhiza* trees that generated forest gaps and promoted the establishment of small trees under 20 ‰ salinity condition.

We did not prescribe any lifespan in the model due to the lack of knowledge on mangroves' longevity (the original SEIB-DGVM defines tree longevity as a model parameter, but this factor was excluded in this study for this reason; the relevant description can be found in L. 100–101 in the Supporting Information). We also did not introduce any specific processes for the death of large trees such as size- and age-dependent mortality (please see Note S3 in the Supporting Information for the processes related to tree mortality). Therefore, tree deaths occur without dependence on tree size or age in the simulation. On the other hand, only deaths of large trees generated forest gaps that promoted tree establishment because the deaths of small trees resulted in growth stimulation of the surrounding trees and the space created by the death were eventually filled by the canopy of the surrounding trees; this process can be seen in Fig. 8 (now Fig. 9) showing the decrease in tree density with increase in individual tree biomass.

In this regard, we figured out that the term "onset of deaths of large *B. gymnorrhiza* trees" in L367 was confusing because it seems as if some specific processes triggered the large trees' mortality. Instead, we should have written it as "onset of formation of forest gaps resulted

from deaths of large *B. gymnorrhiza* trees" to convey our intention correctly. We are sorry for the confusion.

We have corrected this point in the revised manuscript (L. 402, or L. 416 in the marked-up version).

**RC2:**

L394: A figure citation here where this comparison is shown would be helpful.

**Response:**

We have added the figure citation in L. 428 in the revised manuscript or L. 443 in the marked-up version (which is Fig. 8). We thank the reviewer for the suggestion.

**RC2:**

L510: At what time increment is the optimization of the DBH-H adjustment applied?

**Response:**

The morphological adjustment occurs at daily time step through the flexible biomass allocation as indicated in Fig. 2. The detailed procedures on biomass allocation are described in Fig. 3.

**RC2:**

L521: Does the increase in root biomass refer to both above and below ground roots or are the aboveground roots lumped into the shoot category in this scenario?

**Response:**

The statement in L. 521 refers to the result shown in Fig. 9 (now Fig. S7). In the figure, the root biomass refers to the sum of coarse root and fine root biomass while the shoot biomass refers to the sum of stem and leaf biomass. Thus, the above-ground roots are not accounted for in both shoot and root biomass in Fig. 9. The appropriate description has been provided in the caption of Fig. 9 in the original manuscript (L. 422–430).

**RC2:**

L579: It would be nice to see the code released with a DOI in a Zenodo repository or the like given the relevance of this modeling effort to the broader community of models.

**Response:**

We have uploaded the model code in the following link with permission of Dr. Hisashi Sato, the developer of the SEIB-DGVM. We revised the section "Code Availability" correspondingly.

https://github.com/MasayaYoshikai/SEIB_mangrove

**RC2:**
Finally, there are a few instances of minor grammatical errors (subject-verb agreement and plurals versus possessives) that could be addressed through the use of a grammar editing service.

**Response:**
We are sorry about the grammatical errors.
We have carefully checked the manuscript again, and corrected some grammatical errors in the revised manuscript (e.g., L. 17, 47, and 57).

**Response to community comment (CC)**

**CC:**
Figure 7. Comparison of field-measured and modeled (a) mean DBH and (b) AGB of *R. stylosa* and *B. gymnorrhiza* along with soil salinity gradient. How to confirm that the observed pattern is ONLY due to salinity? What about other driving force to make this pattern?

**Response:**
We do not intend to argue that the observed patterns in the forest structures were shaped only by the salinity gradient. Actually, the observed AGB and mean DBH showed some variations even at the same salinity levels, and such variations may have been due to other abiotic and biotic factors.

We thought that the spatial variations in nutrient availability (porewater dissolved inorganic nitrogen concentration, DIN) may explain such variations. However, as shown in the Fig. S5 of the "plot-wise simulation", the spatial variations in DIN did not improve the predictions of the forest structures. Please see L. 548–557 in the original manuscript or L. 560–569 in the revised manuscript for the discussion about this point.

There are some other possible factors that may have affected forest structures (e.g., hydroperiod, historical disturbances, forest age, phosphorus availability), but we cannot, at present, assess those effects because of lack of these information from the site. These effects would be examined in future studies.

Nevertheless, salinity is clearly the major factor that have shaped the forest structural patterns as shown in Fig. 7 (now Fig. 8), and the model predicted well the general patterns observed across the salinity gradient, which is the primary scope of this study.

**CC:**

Figure 6. Temporal dynamics in above-ground biomass (AGB). The scenario (d) shows a low AGB but still showing a reasonable LAI in 100 years simulation. This low AGB doesn't correspond to Figure 5, where vegetation cover indicates a high AGB.

**Response:**

The results shown in Figs. 5 and 6 (now Figs. 6 and 7) are consistent.

First, we assume that this comment is about the model results for *Rhizophora stylosa*.

We think this comment came about because of the axis scaling of Fig. 6, which made it difficult to see the small variations of AGB in short time scale (e.g., 50 years). We provide below the temporal dynamics of AGB, LAI, and mean DBH for 34‰ salinity magnified for the first 200 years (Fig. R5).

This figure shows more clearly the substantial increase in AGB of *R. stylosa* from 50 years to 100 years, which corresponds to the results shown in Fig. 5.

Because Fig. 6 was intended to show long-term dynamics and comparison among different salinity conditions, we would like to keep the axis scaling as it is.

Regarding the comment "The scenario (d) shows a low AGB but still showing a reasonable LAI in 100 years simulation.", we cannot assess the accuracy of the predicted LAI at this point because of the lack of observation/field data at the study site; we would like to consider this as one of the future tasks through monitoring. Nevertheless, we compared the simulated relationship between AGB and LAI with published data from other mangrove forests (Fig. R6). Although these may not be comparable due to the different environmental settings and species, the simulation showed reasonable prediction of the general trend of AGB–LAI relationship. Therefore, we consider that the predicted LAI and AGB are reasonable.

In the revised manuscript, we have included a statement in the discussion section that the simulated LAI has not been validated and will be considered for future study (L. 476–479, or L. 515–517 in the marked-up version).

[Figure]

Figure R5: Temporal dynamics in (a) above-ground biomass (AGB), (b) leaf area index (LAI), and (c) mean diameter at breast height (DBH) of *Rhizophora stylosa* (*R. s*) and *Bruguiera gymnorrhiza* (*B. g*) under 34 ‰ soil salinity (*sal*) condition. Trees with DBH < 0.05 m were not included in the calculation of mean DBH. The results are from a simulation corresponding to the one shown in Figure 5 (or 6 in the revised manuscript).

[Figure]

Figure R6: Simulated yearly trajectory of the relationship between AGB and LAI of *R. stylosa* under soil salinity 34 ‰ for the first 200 years, which corresponds to the results shown in Fig.

R5a-b (red circles). Data from Sharma et al. (2017) (black circles) and Salmo et al. (2013) (triangles) are also shown as reference.

**References**

Bonan, G. B., Williams, M., Fisher, R. A., and Oleson, K. W.: Modeling stomatal conductance in the earth system: linking leaf water-use efficiency and water transport along the soil–plant–atmosphere continuum. Geosci. Model Dev., 7(5), 2193–2222, 2014.

Comley, B. W. and McGuinness, K. A.: Above-and below-ground biomass, and allometry, of four common northern Australian mangroves. Aust. J. Bot., 53(5), 431–436, 2005.

Deshar, R., Sharma, S., Mouctar, K., Wu, M., Hoque, A. T. M. R., and Hagihara, A.: Self-thinning exponents for partial organs in overcrowded mangrove *Bruguiera gymnorrhiza* stands on Okinawa Island, Japan. Forest Ecol. Manag., 278, 146–154, 2012.

Jiang, G. F., Goodale, U. M., Liu, Y. Y., Hao, G. Y., and Cao, K. F.: Salt management strategy defines the stem and leaf hydraulic characteristics of six mangrove tree species. Tree Physiol., 37(3), 389–401, 2017.

Kamara, M., Deshar, R., Sharma, S., Kamruzzaman, M. D., and Hagihara, A.: The self-thinning exponent in overcrowded stands of the mangrove, *Kandelia obovata*, on Okinawa Island, Japan. J. Oceanogr., 68(6), 851-856, 2012.

Khan, M. N. I., Sharma, S., Berger, U., Koedam, N., Dahdouh-Guebas, F., and Hagihara, A.: How do tree competition and stand dynamics lead to spatial patterns in monospecific mangroves?. Biogeosciences, 10(4), 2803–2814, 2013.

Komiyama, A., Poungparn, S., and Kato, S.: Common allometric equations for estimating the tree weight of mangroves. J. Trop. Ecol., 21(4), 471–477, 2005.

Perri, S., Katul, G. G., and Molini, A.: Xylem–phloem hydraulic coupling explains multiple osmoregulatory responses to salt stress. New Phytol., 224(2), 644–662, 2019.

Reef, R. and Lovelock, C. E.: Regulation of water balance in mangroves. Ann. Bot-London, 115(3), 385–395, 2015.

Salmo, S. G., Lovelock, C., and Duke, N. C.: Vegetation and soil characteristics as indicators of restoration trajectories in restored mangroves. Hydrobiologia, 720(1), 1-18, 2013.

Sharma, S., Nadaoka, K., Nakaoka, M., Uy, W. H., MacKenzie, R. A., Friess, D. A., and Fortes, M. D.: Growth performance and structure of a mangrove afforestation project on a former seagrass bed, Mindanao Island, Philippines. Hydrobiologia, 803(1), 359-371, 2017.

Trugman, A. T. anderegg, L. D., Wolfe, B. T., Birami, B., Ruehr, N. K., Detto, M., Bartlett, M. K., and Anderegg, W. R.: Climate and plant trait strategies determine tree carbon allocation to leaves and mediate future forest productivity. Glob. Change Biol., 25(10), 3395–3405, 2019.

Trugman, A. T., Detto, M., Bartlett, M. K., Medvigy, D., Anderegg, W. R. L., Schwalm, C., Schaffer, B., and Pacala, S. W.: Tree carbon allocation explains forest drought-kill and recovery patterns. Ecol. Lett., 21(10), 1552-1560, 2018.

Yoshikai, M., Nakamura, T., Suwa, R., Argamosa, R., Okamoto, T., Rollon, R., Basina, R., Primavera-Tirol, Y. H., Blanco, A. C., Adi, N. S., and Nadaoka, K.: Scaling relations and substrate conditions controlling the complexity of Rhizophora prop root system. Estur. Coast. Shelf S., 248, 107014, 2021.